# FlatLab: A Unified Methodology Framework and Simulation-Based Benchmark for Robotic Manipulation of Flat Objects

Xingyu Zhu [1 2]   Wenshuo Han [1]   Zhouyu Wang [1]   Yuran Wang [3]   Ruihai Wu [3]   Hao Dong [3]   Fan Tang [4]
Hechang Chen [1 2]   Hyung Jin Chang [5]   Yixing Gao [1 2]

## Abstract

Robotic manipulation of flat objects is challenging due to the ungraspable configurations and strong variations in object geometry and material. Existing methods rely on heuristic pre-manipulation and are often evaluated in closed settings with limited generalization. We propose a unified framework that decouples the manipulation into a strategy generator and an action execution module. The strategy generator predicts appropriate manipulation strategies from object point clouds by learning strategy-centric, object-invariant representations via simulated data transformation and contrastive learning. Conditioned on the predicted strategy, the execution module decomposes long-horizon manipulation into reusable action primitives and dynamically composes them to generate stable trajectories. To enable systematic evaluation, we introduce FlatLab, a comprehensive simulation benchmark for robotic flat object manipulation. FlatLab provides high-fidelity physical simulation of diverse rigid and deformable flat objects, automated multimodal data collection, and standardized task definitions and evaluation protocols. Experiments conducted in FlatLab demonstrate that our approach generalizes effectively to unseen objects and categories, outperforming existing baselines. The project page and the code are provided at https://flatlab-web.github.io/.

## 1. Introduction

Flat objects, such as magazines and wooden boards, are ubiquitous in everyday life (Kappler et al., 2012; Lévesque et al., 2018; Wu et al., 2023). Accurate robot grasping and manipulation of flat objects is a crucial capability for embodied intelligence (Chen et al., 2023; Ding et al., 2024). Grasping flat objects is challenging because they often offer no readily accessible grasp affordance, such as a book or cloth lying flush on a table (Eppner et al., 2015; Hang et al., 2019; Zhou & Held, 2023). Some studies have developed specialized end-effectors such as dexterous hands and exploited friction to grasp flat objects (Nahum & Sintov, 2022; Jiang et al., 2025; Tran et al., 2025). These advances have pushed the hardware frontier, yet progress is now hampered by algorithmic bottlenecks and high training costs. With conventional two-finger grippers, many studies have pursued pre-manipulation strategies that reconfigure flat objects into graspable poses (Zhang et al., 2023; Kim et al., 2023; Wang et al., 2025a). Typical examples include pushing objects to table edges to enable side grasps, or exploiting nearby vertical surfaces to tilt objects into graspable configurations (Liang et al., 2021; Wang et al., 2024a; Mao et al., 2025). Recently, (Wang & Kasaei, 2025) have proposed dual-arm lifting of large flat objects. These studies have significantly advanced robotic flat object manipulation.

In this paper, we tackle the generalization of flat object manipulation across diverse shapes and materials, eschewing the confines of closed datasets and single-strategy pipelines. As illustrated in Figure 2, no single manipulation strategy is sufficient for all flat objects. Pushing objects to table edges works well for moderately sized rigid items, but fails for large, thick, or deformable ones. Dual-arm lifting requires sufficient object thickness and becomes unreliable for thin or deformable objects. To address these limitations, we propose a unified methodology framework that decouples strategy selection from action execution. The framework comprises two modules: a manipulation strategy generator and a robot action execution module. The strategy generator predicts a manipulation strategy from object point clouds by learning strategy-centric, object-invariant representations through simulated data transformation and con-

---

[1]Jilin University, Changchun, China [2]Engineering Research Center of Knowledge-Driven Human-Machine Intelligence, MoE, Changchun, China [3]Peking University, Beijing, China [4]Chinese Academy of Sciences, Beijing, China [5]University of Birmingham, Birmingham, UK. Correspondence to: Yixing Gao <gaoyixing@jlu.edu.cn>.

*Proceedings of the 43$^{rd}$ International Conference on Machine Learning*, Seoul, South Korea. PMLR 306, 2026. Copyright 2026 by the author(s).

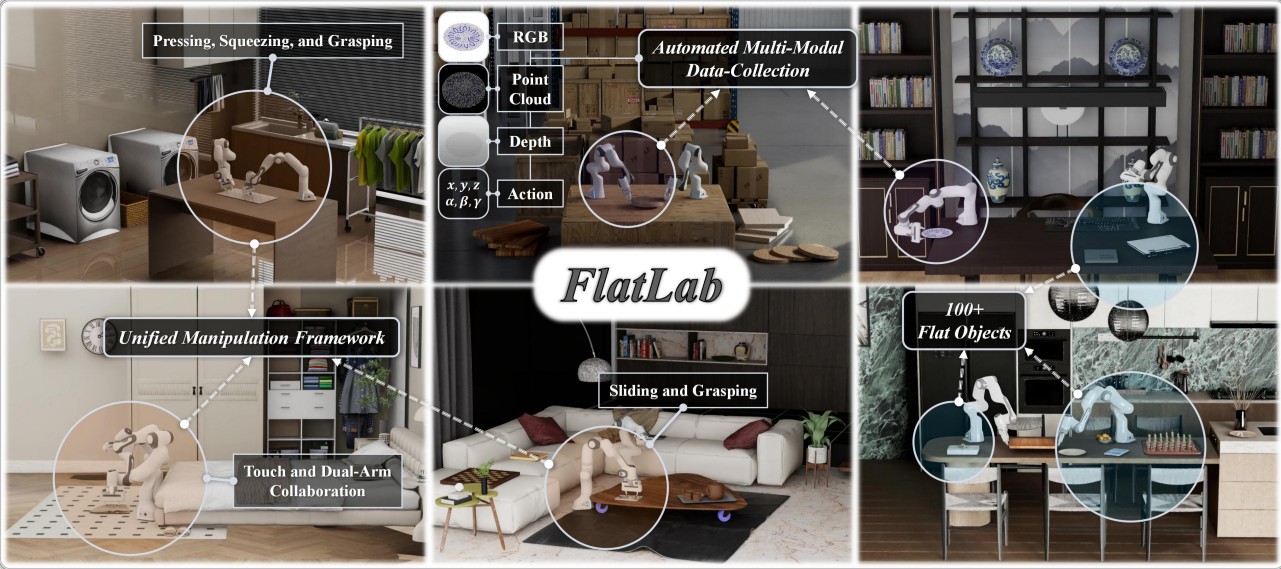

*Figure 1.* **Overview of FlatLab: a comprehensive simulation platform for robotic flat object manipulation.** The platform offers high-fidelity physical simulation of over 100 flat objects and configurable manipulation scenarios. It includes our unified manipulation framework and supports automated multi-modal data collection, standardized task definitions, unified evaluation protocols, and turnkey deployment scripts. All functionalities are fully reconfigurable and extensible, facilitating ongoing research in flat object manipulation.

trastive learning. The execution module then decomposes long-horizon manipulation into reusable action primitives and dynamically composes them to generate smooth and reliable manipulation trajectories. This decoupled design prevents overfitting to object-specific geometry and enables robust generalization to unseen objects and categories.

Beyond the algorithmic challenge, the lack of a standardized benchmark has also hindered progress in flat object manipulation. Recent work in embodied intelligence increasingly uses large-scale simulation benchmarks to evaluate manipulation performance and generalization (Xiang et al., 2020; Xian et al., 2023; Wang et al., 2025b; Li et al., 2025). (Xian et al., 2023) proposed FluidLab for benchmarking complex fluid manipulation. (Lu et al., 2024) released GarmentLab for robotic clothing manipulation. (Li et al., 2025) designed LabUtopia for laboratory tasks. However, despite the prevalence of flat objects in real-world environments, no unified simulation benchmark exists for robotic flat object manipulation that systematically covers rigid and deformable objects.

To fill this gap, we introduce FlatLab as shown in Figure 1, a comprehensive simulation platform for robotic flat object manipulation built on Isaac Sim (NVIDIA, 2023). FlatLab provides over 100 physically simulated flat objects spanning rigid and deformable categories, along with configurable manipulation scenarios and standardized task definitions. The platform supports automated multi-modal data collection, including point clouds, RGB images, depth images, and action demonstration data. Unified evaluation protocols and one-click deployment scripts further facilitate reproducible experimentation. FlatLab's functionalities are reconfigurable and extensible, facilitating continued research.

We conduct extensive experiments in FlatLab to evaluate our unified framework, and the results demonstrate its effectiveness. In particular, generalization tests and baseline comparisons show superior performance on unseen objects and categories. Extensive ablation studies also demonstrate the effectiveness of each module and parameter in our method. Our contributions can be summarized as follows:

- We propose a unified robotic flat object manipulation framework that decouples manipulation strategy prediction from action execution, enabling robust and generalizable flat object grasping across unseen objects and categories.

- We introduce FlatLab, a comprehensive simulation platform for robotic flat object manipulation, featuring high-fidelity physical simulation, automated multi-modal data collection, standardized tasks, and unified evaluation protocols.

- Extensive experiments in FlatLab demonstrate superior generalization performance on unseen objects and categories, supported by thorough baseline comparisons and ablation studies.

## 2. Related Work

### 2.1. Robotic Manipulation of Flat Objects

Enabling robots to grasp flat objects has always been a challenging task (Kappler et al., 2012; Lévesque et al., 2018; Chen et al., 2023; Wu et al., 2023; Ding et al., 2024). This

*Figure 2.* **Motivation of Unified Flat Object Manipulation.** Different flat objects exhibit distinct failure modes under single-strategy pipelines: edge-pushing fails for large or deformable objects, while dual-arm lifting is unreliable for thin or compliant ones. These challenges motivate a unified approach that adaptively selects strategies and executes actions for diverse flat objects.

is because objects lying flush against a table, such as horizontal magazines, keyboards, or fabric sheets, provide few accessible grasp positions (Eppner et al., 2015; Hang et al., 2019; Kim et al., 2023; Zhou & Held, 2023; Wang et al., 2025a). Some studies have addressed the problem by designing specialized end-effectors such as dexterous hands or flexible grippers (Tong et al., 2020; Zhang et al., 2022; Nahum & Sintov, 2022; Jiang et al., 2025; Tran et al., 2025). Other studies propose pre-manipulation strategies to create graspable positions for flat objects, such as sliding a book to the table edge or relying on the auxiliary force of a wall (Sun et al., 2020; Liang et al., 2021; Zhang et al., 2023; Wang et al., 2024a; Mao et al., 2025). Recently, (Wang & Kasaei, 2025) has proposed dual-arm lifting of large flat objects. While these approaches have advanced flat object manipulation, they are typically designed around a single manipulation strategy and evaluated on closed object sets, limiting generalization across diverse shapes and materials. In this paper, we focus on strategy-level generalization for flat object grasping by unifying multiple complementary manipulation strategies within a unified framework and predicting the appropriate strategy from object point clouds.

### 2.2. Simulation Benchmark for Robotic Manipulation

Recently, rapid advances in simulation technology have enabled robotic manipulation platforms to achieve significant progress (Huang et al., 2021; Xian et al., 2023; Wang et al., 2024b). Gazebo (Koenig & Howard, 2004), MuJoCo (Todorov et al., 2012), and PyBullet (Coumans & Bai, 2016) support real-time physics simulation and the integration of deep-learning pipelines. Some research integrates fully functional physics engines and supports a variety of reinforcement-learning tasks (Brockman et al., 2016; Duan et al., 2016; Tassa et al., 2018; Urakami et al., 2019; James et al., 2020). Others focus on human-robot interaction in simulated environments (Savva et al., 2019; Zheng et al., 2022; Gong et al., 2023; Tao et al., 2024). LabUtopia (Li et al., 2025) allows embodied agents to learn manipulation, navigate, and tasks planning in simulated laboratories. Despite these advances, there is still no unified simulation platform dedicated to robotic flat object manipulation that

systematically covers rigid and deformable objects. To fill this gap, we introduce FlatLab, a simulation-based benchmark built on Isaac Sim (NVIDIA, 2023) that supports high-fidelity physical simulation of diverse flat objects, automated multi-modal data collection, and standardized task definitions and evaluation protocols. A comparison with existing simulation platforms is provided in Appendix B.

## 3. Unified Robotic Flat Object Manipulation

We construct a unified framework illustrated in Figure 3, which enables robotic grasping of diverse flat objects. As illustrated in Figure 2, the first challenge lies in adapting manipulation strategies to the diverse shapes and materials of flat objects. To address this, we introduce three complementary strategies that together cover various flat objects: **Strategy-A** (pushing the object to the table edge, for thin and small objects, such as disks and pads); **Strategy-B** (co-operative lifting with two arms for thick and large objects, such as boxes and paintings); and **Strategy-C** (squeezing the edges to create graspable folds for deformable objects, such as towels and fabrics). The model then learns shape and material representations of the manipulated objects and establishes an adaptive mapping from features to strategies through simulated data transformation and contrastive learning, enabling cross-strategy generalization.

The second challenge is achieving accurate long-horizon manipulation of unseen objects and categories with low-cost data, which remains a key difficulty for current end-to-end large vision-language-action models and diffusion policy (Chi et al., 2023; Moo et al., 2025). To address this, we decouple the manipulation into two modules: the **Manipulation Strategy Generator** for predicting the manipulation strategy, and the **Robot Action Execution Module** for executing the corresponding robot actions. Simultaneously, we formulate long-horizon manipulation as a sequence of action primitives. These primitives avoid high data-collection costs while ensuring stable long-horizon manipulation. The independent nature of strategies and action primitives prevents the model from overfitting to the training set, allowing it to effectively generalize to unseen objects and categories.

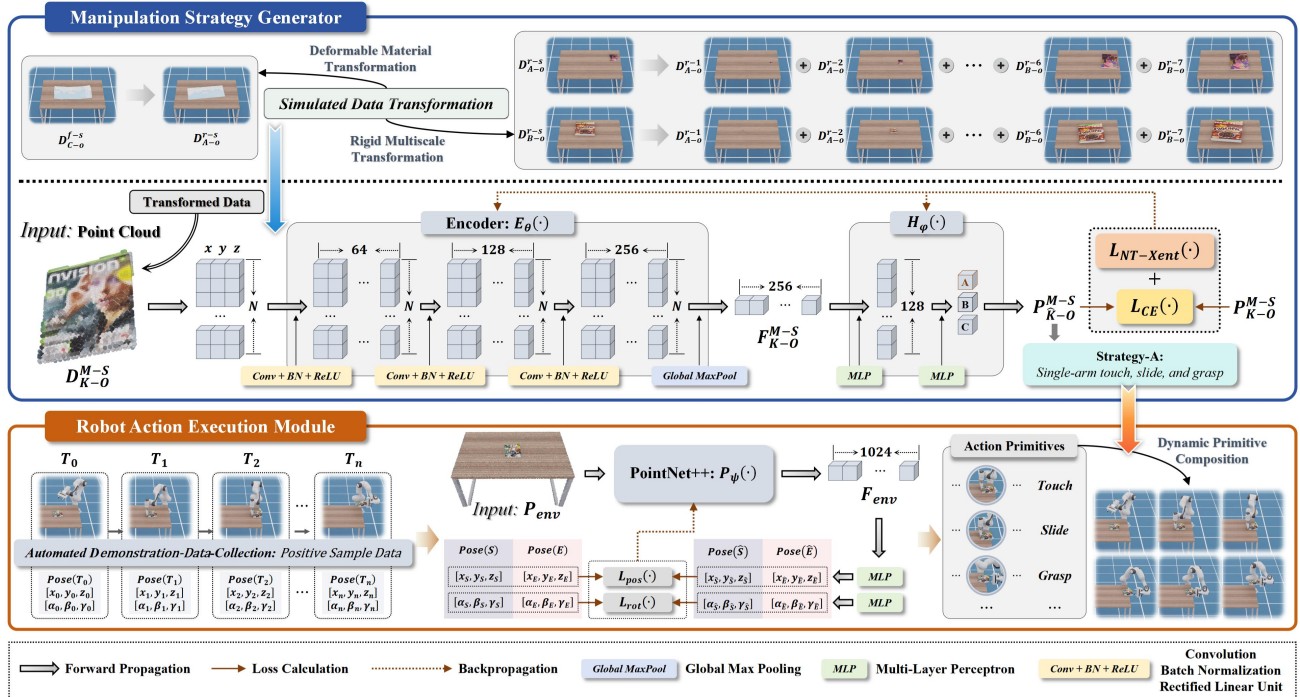

*Figure 3.* **Unified Robotic Flat Object Manipulation Framework.** The Manipulation Strategy Generator takes point clouds of flat objects as input, embeds their features into a strategy-centric representation via multi-view consistency contrastive learning, and then predicts the manipulation strategy. The predicted strategy and the scene point cloud are fed to the Robot Action Execution Module. This module decomposes long-horizon manipulation into reusable action primitives, learns position-dependent poses rather than object-specific features, and then sequences the primitives to produce smooth trajectories.

## 3.1. Problem Formulation

Figure 3 illustrates the pipeline of our framework. Specifically, the entire framework consists of a manipulation strategy generator and a robot action execution module, defined as $Generator$ and $Execution$ respectively. The input of the manipulation strategy generator is the point cloud of the flat object, defined as $D_{K-O}^{M-S}$, and the output is the manipulation strategy, defined as $P_{\hat{K}-O}^{M-S}$. $M = \{r, f\}$ represents the object material, including rigidity ($r$) and deformability ($f$). $K = \{A, B, C\}$ represents the manipulation strategies, including Strategy-A ($A$), Strategy-B ($B$), and Strategy-C ($C$). $S = \{s, 1, 2, 3, 4, 5, 6, 7\}$ represents the object scale, including the original scale ($s$) and rescaled instances ($1, 2, 3, 4, 5, 6, 7$). $O$ represents the object index number. The action execution module receives the scene point cloud $P_{env}$ together with the strategy $P_{\hat{K}-O}^{M-S}$ produced by the manipulation strategy generator, and outputs the dynamic primitive composition $D_{pc}$. The above process can be formalized as follows:

$$Generator(D_{K-O}^{M-S}) \Rightarrow P_{\hat{K}-O}^{M-S}, \quad (1)$$

$$Execution(P_{env}, P_{\hat{K}-O}^{M-S}) \Rightarrow D_{pc}. \quad (2)$$

This formulation decouples perception, strategy reasoning, and action execution into a structured pipeline, enabling efficient and generalizable learning.

## 3.2. Manipulation Strategy Generator

The manipulation strategy generator takes object point clouds as input and predicts an appropriate strategy. Strategy-A (pushing the object to the table edge) is suitable for smaller, thinner objects; strategy-B (cooperative lifting with two arms) is suitable for larger, thicker objects; and strategy-C (squeezing the edges to create graspable folds) is suitable for deformable objects. These strategies enable the grasping of various flat objects and are selected dynamically according to object shape and material properties.

### 3.2.1. BASIC COMPONENT CONSTRUCTION

The object point cloud $D_{K-O}^{M-S}$ is converted into a color-independent tensor and then supplied to the encoder $E_\theta(\cdot)$. $N$ denotes the number of points in the sampled point cloud. The encoder first applies a series of shared point-wise transformations, implemented as multi-layer convolutions, batch-normalization layers, and ReLU activations. The point features are then aggregated via global max pooling to yield a compact global representation, based on which the encoder produces an intermediate feature $F_{K-O}^{M-S}$. Finally, a prediction head $H_\varphi(\cdot)$, implemented as a two-layer MLP, maps the intermediate feature to the logit $P_{\hat{K}-O}^{M-S}$. The forward propagation process can be formalized as follows:

$$E_\theta(D_{K-O}^{M-S}) \Rightarrow F_{K-O}^{M-S}; \quad H_\varphi(F_{K-O}^{M-S}) \Rightarrow P_{\hat{K}-O}^{M-S}. \quad (3)$$

The ground-truth strategy label is denoted by $P_{K-O}^{M-S}$, and the model is trained by minimizing the cross-entropy loss:

$$\min_{\theta,\varphi} |L_{CE}\{H_\varphi[E_\theta(D_{K-O}^{M-S})], P_{K-O}^{M-S}\}|. \quad (4)$$

### 3.2.2. SIMULATED DATA TRANSFORMATION

Accurate prediction of manipulation strategies depends on the effective extraction of the object's shape and material characteristics. The object features corresponding to different manipulation strategies should be decoupled, while those corresponding to the same strategy should be aggregated. Therefore, the model's ability to generalize to unseen objects and categories lies in optimizing multi-view strategy consistency rather than object specific information. To this end, the dataset is first augmented by exploiting the transformability of simulated data, including material and scale transformations. Material transformation yields a rigid counterpart of a deformable object by adjusting the physical parameters of the simulated asset, such as from $D_{C-o}^{f-s}$ to $D_{A-o}^{r-s}$. Consequently, the associated manipulation strategy is updated. Scale transformation first determines the minimum scale, the smallest single-layer point cloud, and the maximum scale that still fits within the desktop size limit. Uniform sampling is then performed between these limits, generating a scale set such as $\{D_{B-o}^{r-i}\}_{i=1}^7$.

### 3.2.3. MULTI-PERSPECTIVE STRATEGY CONSISTENCY

Multi-view strategy consistency optimization is achieved via contrastive learning. Positive pairs share the same strategy label but come from different objects; negative pairs come from the same object yet bear different strategy labels. For instance, given object $D_{K-O}^{M-S}$, the positive samples are $D_{K-\overline{O}}^{M-S}$, and the negative samples are $D_{\overline{K}-O}^{M-S}$, where $\overline{K}$ indicates any strategy other than $K$ and $\overline{O}$ any object other than $O$.

We compute the cosine similarity between the corresponding embedding vectors and optimize with the temperature-scaled NT-Xent loss (Chen et al., 2020). Let $z_{K-O}^{M-S}$ denote the normalized embedding of $D_{K-O}^{M-S}$. The NT-Xent loss is:

$$L_{NT-Xent} = -\log \frac{\exp[(z_{K-O}^{M-S})^\top z_{K-\overline{O}}^{M-S}/\tau]}{\exp[(z_{K-O}^{M-S})^\top z_{\overline{K}-O}^{M-S}/\tau]}, \quad (5)$$

where $\tau$ is the temperature coefficient. This design encourages strategy consistent yet object invariant representations and prevents the network from merely memorizing object specific geometry.

The overall training objective is:

$$\min_{\theta,\varphi} |L_{Generator} = L_{CE} + \lambda L_{NT-Xent}|, \quad (6)$$

where $\lambda$ controls the relative weight of the contrastive regularization. The sensitivity analysis of the relevant hyperparameters is provided in Appendix E.

### 3.3. Robot Action Execution Module

The robot action execution module takes the current scene point cloud and the predicted manipulation strategy as input and outputs the 6-D pose of the end-effector. For flat object grasping, long-horizon pre-manipulation is unstable, data-intensive, and likely to overfit the training data. To meet this challenge and generalize to unseen objects and categories, the module decomposes the long-horizon strategy into action primitives such as touching, sliding, and squeezing. Each primitive is defined as a single path plan specified by its start and end gripper poses. This design lets the model focus on position-dependent staged actions without overfitting to the training objects, thus achieving better generalization.

Each action primitive is learned from the corresponding staged pose and the scene point cloud. Demonstrations are automatically collected with the built-in function of FlatLab, and positive samples are drawn as described in Section 4.3. The collected action data can be defined as a pose $Pose(T_n)$, where $T_n$ denotes the current time step. The scene point cloud encoder $P_\psi(\cdot)$, based on PointNet++ (Qi et al., 2017), yields a feature tensor $F_{env}$. Two parallel MLPs respectively map $F_{env}$ to the 3-D position and the rotation component of the 6-D pose. This decoupling stabilizes optimization and enables specialized losses for translation and rotation. Let the start and end poses be $Pose(S)$ and $Pose(E)$, respectively. The forward pass is:

$$P_\psi(P_{env}) \Rightarrow F_{env}; \quad MLP(F_{env}) \Rightarrow Pose(\cdot). \quad (7)$$

A geometrically consistent pose loss supervises the network. This loss comprises a translation term $L_{pos}(\cdot)$ and a rotation term $L_{rot}(\cdot)$. Given the normalized predicted position $\hat{\mathbf{p}}$ and the ground-truth position $\mathbf{p}$, the translation loss is:

$$L_{pos} = \|\hat{\mathbf{p}} - \mathbf{p}\|_2^2. \quad (8)$$

Because the quaternion double covers $SO(3)$ and must remain unit, we employ the geodesic distance. Given the normalized predicted quaternion $\hat{\mathbf{q}}$ and the ground-truth quaternion $\mathbf{q}$, the angular error and the rotation loss are:

$$\theta = 2\arccos(|\langle\hat{\mathbf{q}}, \mathbf{q}\rangle|); \quad L_{rot} = \mathbb{E}[\theta^2]. \quad (9)$$

This provides smooth gradients and physically meaningful supervision.

The final training objective is:

$$\min_{\psi} |L_{Execution} = \lambda_{pos}L_{pos} + \lambda_{rot}L_{rot}|, \quad (10)$$

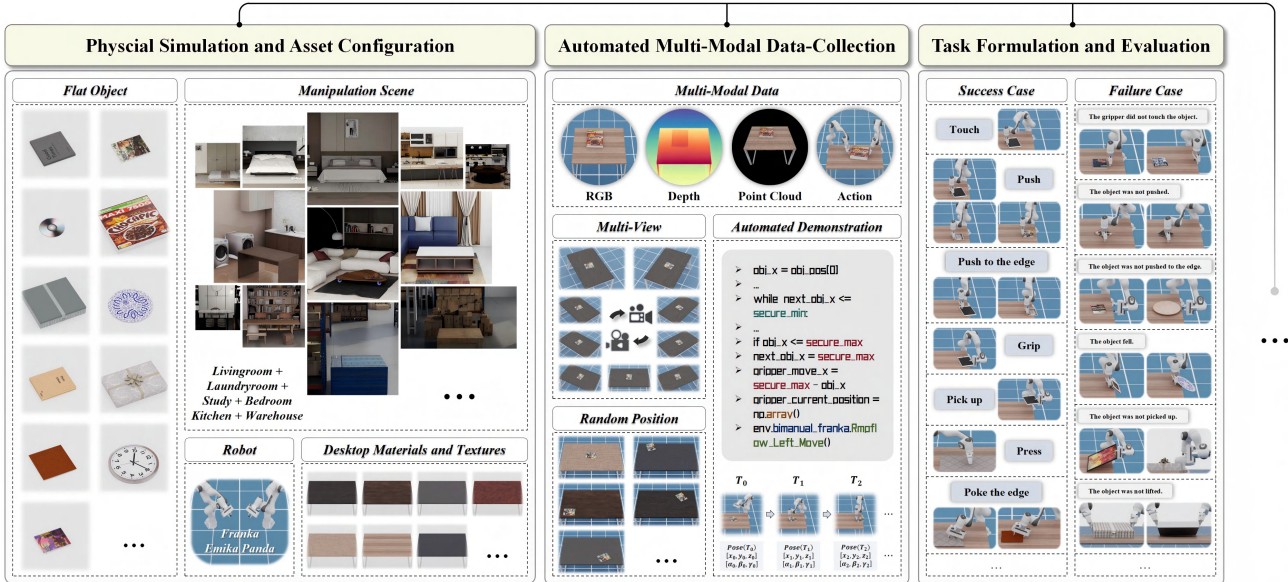

*Figure 4.* **Main Function Modules of FlatLab.** First, FlatLab provides configurable assets with high-precision physical simulations, including flat objects, manipulation scenarios, and associated material textures. Second, FlatLab supports automated multi-modal data collection that yields perceptual and kinematic data from both successful and failed demonstrations. Third, FlatLab formalizes a set of robotic flat object manipulation tasks together with their evaluation criteria. FlatLab's functionalities are fully reconfigurable and extensible, facilitating continued research.

where $\lambda_{pos}$ and $\lambda_{rot}$ balance translation and rotation errors. The trained primitives are dynamically invoked and sequenced according to the predicted strategy to generate a smooth trajectory, enabling reliable manipulation and grasping of flat objects. This strategy driven method enables the learned primitives to adapt to different object geometries and materials, thereby achieving stable and generalized operational performance. The sensitivity analysis of the relevant hyperparameters is provided in Appendix E.

## 4. Simulation-Based Benchmark of FlatLab

We have developed FlatLab, a robot flat object manipulation simulation platform. Figure 4 shows a block diagram of the functionalities. The platform features high-precision, configurable physical simulation assets, including over 100 flat objects, more than 20 manipulation scenarios, and over 10 desktop materials and textures. It provides automated multi-modal data-collection capabilities, with built-in configurable multi-view cameras, automatic generation of manipulation demonstrations. The platform also standardizes various tasks for simulation evaluation and supports model integration and replacement. The functionality is also reconfigurable and extensible, supporting ongoing research. The details of FlatLab are provided in Appendix A.

### 4.1. Environment Setup

FlatLab is built on NVIDIA Isaac Sim 4.5.0 (NVIDIA, 2023). The Isaac Sim engine provides a highly parallel

data pipeline, realistic rendering, multi-sensor simulation, and seamless integration with the Robot Operating System (ROS). We use the RMPFlow (Cheng et al., 2021) controller for action planning.

### 4.2. Physical Simulation and Asset Configuration

We have independently developed and integrated more than 100 high-precision assets for flat object physical simulation, spanning 21 categories such as Book, Box, and Fabric, each with configurable attributes. Each asset is labeled with empirically derived physical properties such as color, texture, mass, and friction coefficient. We use the Universal Scene Description (USD) file to store all assets and their physical properties, semantic information, and rendering parameters. We have implemented accurate contact and collision for rigid objects. For deformable objects, we use GPU-accelerated Position-Based Dynamics (PBD) (NVIDIA, 2023) to simulate interactions efficiently. We also provide 18 manipulation scenarios across 6 categories, including **Bedroom**, **Kitchen**, **Laundryroom**, **Livingroom**, **Study** and **Warehouse**, with different desktop backgrounds with varied materials and textures. We use two 7-degree-of-freedom Franka Emika Panda robotic arms equipped with parallel grippers, supporting both single-arm and dual-arm manipulation. Isaac Sim's built-in motion planner maps end-effector commands to joint-space trajectories for execution. Robot simulation achieves precise force control, position-velocity control, and inverse dynamics through the PhysX joint system (NVIDIA, 2023). We build complete simula-

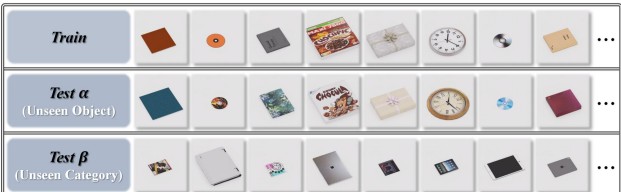

*Figure 5.* **Flat Object Dataset.** The dataset is partitioned into three splits: a training set (***Train***), a test set with unseen objects from the same categories (***Test*** $\alpha$), and a test set with objects from unseen categories (***Test*** $\beta$). This partition enables a comprehensive evaluation of both object-level and category-level generalization.

tion scenes by importing individual mesh models and URDF (Unified Robot Description Format) files for the robots.

### 4.3. Automated Multi-Modal Data-Collection

Collecting demonstration data manually, whether by teleoperation or inverse kinematics, is slow and labor intensive, so automation is essential. FlatLab offers automated multi-modal data collection that yields perceptual data such as point clouds, RGB images and depth images from cameras, and kinematic data such as joint positions and end-effector 6D poses. Users can adjust camera viewpoints and randomized object placements through a single parameter set. The automation script samples randomized robot configurations, labels each trial as success or failure against quantified task criteria, and compiles a demonstration dataset from which any subset of modalities can be selected.

During data collection, the system can be configured to operate in headless mode, increasing efficiency by at least a factor of two. A centralized simulation controller is used to initialize the physical environment, manage simulation steps, and synchronize sensor rendering and data recording. We have implemented two data collection flows, one for rigid objects and one for deformable objects. Although these two flows handle objects with different physical properties, they share a unified execution flow with steps including environment reset, object instantiation, scene randomization, physical stabilization, and sensor based observation data collection. This design guarantees data format consistency across different object types.

### 4.4. Task Formulation and Evaluation

FlatLab defines a comprehensive set of tasks for evaluating robot manipulation of flat objects in simulation. The set includes single-arm touching, sliding on the tabletop, sliding to the table edge, grasping, picking up, dual-arm touching, cooperative lifting, pressing, squeezing the edge, and grasping the edge, and can be applied to any configurable asset. Users can evaluate agents on individual tasks or chain them into extended scenarios such as tidying objects to match human preferences and conducting human-robot interaction. The proposed unified framework for grasping flat objects

is provided as a baseline in FlatLab. We adopt success rate as the main metric and provide clear success and failure conditions. The success criterion requires that the object stay aloft without slipping for a predefined interval after the final action is executed. Following prior work, we set this interval to two seconds. Additionally, FlatLab supports evaluation under various environmental conditions, encompassing variations in both scene configuration and object states. valuation metrics extend beyond success rate to include task efficiency, trajectory smoothness, and primitive action count, enabling fine-grained agent performance analysis. We will publicly release the entire FlatLab project, offering a reconfigurable and extensible platform for robotic flat object manipulation.

## 5. Experiments

### 5.1. Dataset

We first constructed a dataset for training and evaluating the framework. We used all 104 flat objects from 21 categories available in FlatLab. As shown in Figure 5, these objects were partitioned into a **Train** set, a **Test** $\alpha$ set, and a **Test** $\beta$ set. **Test** $\alpha$ set contains objects from the same categories as **Train** set but never seen during training. **Test** $\beta$ set comprises objects from entirely new categories. Each object was annotated with the corresponding manipulation strategy. Using FlatLab's automated multi-modal data-collection pipeline, we generated separate datasets for the Manipulation-Strategy Generator and the Action-Execution Module. For the Manipulation-Strategy Generator, we randomized object placements and applied simulated data transformation, yielding 1390 object point clouds. For the Action-Execution Module, we recorded 50 demonstrations per object, each consisting of a scene point cloud and the corresponding 6-DoF gripper pose. More details are provided in Appendix C.

### 5.2. Robotic Manipulation Evaluation

Our method achieved strategy discrimination accuracies of **99.2%**, **91.3%**, and **78.6%** on the **Train**, **Test** $\alpha$, and **Test** $\beta$ sets, respectively. Our robot grasping experiments were conducted in a simulated desktop environment with randomized object positions to mimic variations in real-world settings. Each object underwent 5 grasping trials in different orientations. We evaluated grasping performance on all 104 flat objects. Following the FlatLab evaluation benchmark, we achieved grasping success rates of **81.1%**, **74.2%**, and **69.0%** on the **Train**, **Test** $\alpha$, and **Test** $\beta$ sets, respectively. Experimental results demonstrate the effectiveness of the proposed framework. Moreover, the results on the **Test** $\alpha$ and **Test** $\beta$ sets indicate strong generalization ability. Figure 6 illustrates some examples of the robot's flat object grasping process. More details are provided in Appendix D.

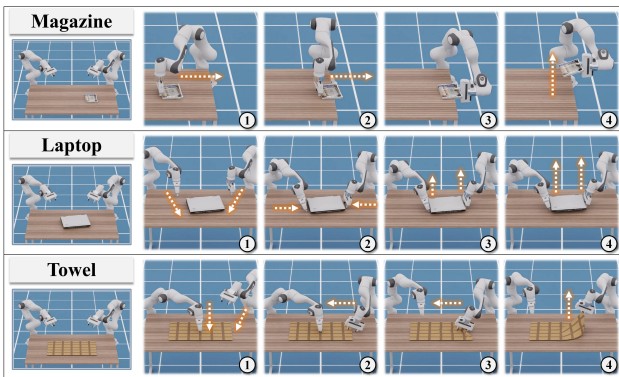

*Figure 6.* **Robotic Grasping Evaluation.** Our method adaptively selects strategies to enable the grasping of flat objects of various shapes and materials, demonstrating significant generalizability.

## 5.3. Comparison with Baselines

We constructed multiple baselines for comparative experiments to demonstrate the superiority and generalization ability of our proposed framework. The first baseline, **Slide**, employs a strategy of sliding the object to the edge of the table and then performing a grasp. This baseline was inspired by (Ding et al., 2024). The second baseline, **Lift**, adopts a strategy that combines dual-arm coordination and end-effector friction to lift flat objects. This baseline was inspired by (Wang & Kasaei, 2025). **Diffusion Policy** (Chi et al., 2023), **3D Diffusion Policy** (Ze et al., 2024), **OpenVLA** (Kim et al., 2025), $\pi_0$ (Moo et al., 2025), and $\pi_{0.5}$ (Black et al., 2025) are trained end-to-end to highlight the benefits of decoupling strategy from execution and of invoking action primitives. Following the FlatLab evaluation benchmark, we present quantitative comparison results in Table 1. For each baseline, we conducted 5 grasping trials for each flat object. The average success rate of grasping all flat objects was used as the evaluation metric. Experimental results indicate that the baseline methods are limited either by a single manipulation strategy or by overfitting to the training set. Our approach dynamically selects manipulation strategies based on object shape and material, thereby achieving cross-category generalization.

This advantage mainly comes from the decoupled framework design, where the system first determines the manipulation strategy and then executes the corresponding action primitive. Compared with end-to-end policies that directly predict actions, the proposed framework focuses more on learning strategy-level representations instead of memorizing object-specific patterns. In addition, simulated data augmentation and contrastive learning further improve the robustness and generalization capability of the model under limited training data. Moreover, flat object manipulation requires fine-grained decision making to maintain stable contact with the tabletop while simultaneously creating graspable positions. Under such conditions, constraining the manipulation process with explicit gripper poses becomes

*Table 1.* **Quantitative Comparison Results.** Our method consistently outperforms all baselines on both training and test sets.

| Method | Train | Test $\alpha$ | Test $\beta$ |
|---|---|---|---|
| **Slide** | 32.3% | 27.3% | 24.7% |
| **Lift** | 41.2% | 35.0% | 27.2% |
| **Diffusion Policy** | 48.8% | 36.8% | 38.8% |
| **3D Diffusion Policy** | 66.2% | 54.2% | 51.2% |
| **OpenVLA** | 55.5% | 22.6% | 17.8% |
| $\pi_0$ | 63.0% | 23.6% | 17.3% |
| $\pi_{0.5}$ | 68.6% | 28.6% | 25.0% |
| **FlatLab (Ours)** | **81.1%** | **74.2%** | **69.0%** |

particularly beneficial. A representative example is the manipulation of the Disk object in FlatLab. For extremely thin objects, our method can accurately push the object to the table edge before grasping. In contrast, VLA-based methods may suffer from action jitter and discontinuity, which can cause the gripper to collide with or rub against the tabletop, or even fail to establish contact with the object. In particular, we propose a novel strategy for deformable flat objects, a scenario not addressed in previous studies. More details for comparisons are provided in Appendix F.

## 5.4. Ablation Studies

We conducted extensive ablation experiments to analyze the contribution of each key component in the proposed unified grasping framework. All experiments were evaluated on three datasets: **Train**, **Test** $\alpha$, and **Test** $\beta$. We report the strategy discrimination accuracy of the Manipulation Strategy Generator and the grasp position prediction error of the Robot Action Execution Module. Quantitative results are summarized in Table 2, where **A-1** to **A-8** correspond to different ablation configurations. Overall, removing any of the major components leads to a noticeable performance drop, confirming that the proposed framework benefits from the joint design of data transformation, contrast consistency, primitive decomposition and rotation loss. More analysis of ablation experimental results are provided in Appendix G.

## 5.5. Real-World Evaluation

To demonstrate real-world manipulation performance, we deployed our method on a Baxter robot and evaluated its performance. Our real-world experimental setup is illustrated in Figure 7. The camera's field of view covers the entire tabletop. We employ the Segment Anything Model (SAM) (Kirillov et al., 2023) to distinguish objects from the tabletop, enabling the extraction of data for each individual object. We collected more than 30 flat objects from the real world, split them into training and test sets, and mirrored the simulation pipeline and evaluation protocol exactly. On the real-world data, we achieved average grasping success rates

*Table 2.* **Ablation study results of the proposed unified robotic flat object grasping framework.** The Manipulation Strategy Generator and the Robot Action Execution Module are evaluated separately through ablation. Strategy Accuracy denotes the strategy discrimination accuracy of the Manipulation Strategy Generator, while Grasping P-MSE denotes the prediction error of the grasp pose position in the Robot Action Execution Module. **A-1** to **A-8** correspond to different ablation configurations.

| | | A-1 | A-2 | A-3 | A-4 | A-5 | A-6 | A-7 | A-8 | Full |
|---|---|---|---|---|---|---|---|---|---|---|
| Generator Input | **RGB** | | | | | | | ✓ | | |
| | $P_{obj}$ | ✓ | ✓ | ✓ | ✓ | ✓ | ✓ | | | ✓ |
| | $P_{env}$ | | | | | | | | ✓ | |
| Data Transformation | | ✓ | | | ✓ | ✓ | ✓ | ✓ | ✓ | ✓ |
| Contrast Consistency | | | ✓ | | ✓ | ✓ | ✓ | ✓ | ✓ | ✓ |
| Primitive Decomposition | | ✓ | ✓ | ✓ | ✓ | | | ✓ | ✓ | ✓ |
| Rotation Loss | | ✓ | ✓ | ✓ | | ✓ | | ✓ | ✓ | ✓ |
| Strategy Accuracy | **Train** | 97.1% | 99.2% | 99.2% | **99.2%** | **99.2%** | **99.2%** | 94.8% | 85.7% | **99.2%** |
| | **Test** $\alpha$ | 88.0% | 91.3% | 90.7% | **91.3%** | **91.3%** | **91.3%** | 82.3% | 79.3% | **91.3%** |
| | **Test** $\beta$ | 77.3% | 71.4% | 70.5% | **78.6%** | **78.6%** | **78.6%** | 56.4% | 70.5% | **78.6%** |
| Grasping P-MSE | **Train** | **0.0032** | **0.0032** | **0.0032** | 0.0052 | 0.0081 | 0.0102 | **0.0032** | **0.0032** | 0.0032 |
| | **Test** $\alpha$ | **0.0034** | **0.0034** | **0.0034** | 0.0062 | 0.0083 | 0.0107 | **0.0034** | **0.0034** | 0.0034 |
| | **Test** $\beta$ | **0.0040** | **0.0040** | **0.0040** | 0.0077 | 0.0076 | 0.0114 | **0.0040** | **0.0040** | 0.0040 |

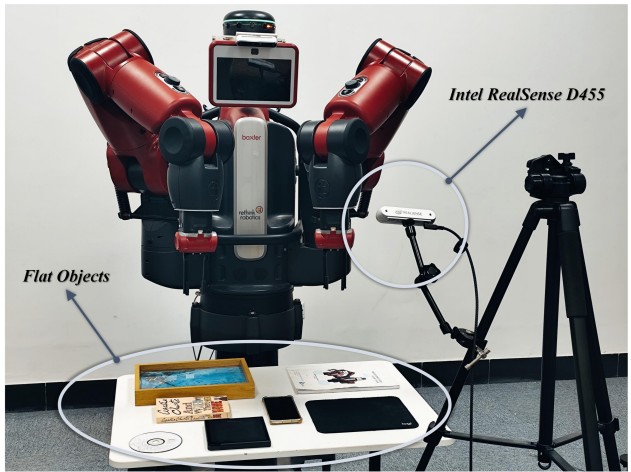

*Figure 7.* **Real-World Experimental Scenario Setup.** The experimental setup consists of a Baxter dual-arm robot, a table positioned directly in front of the robot, and an Intel RealSense D455 camera.

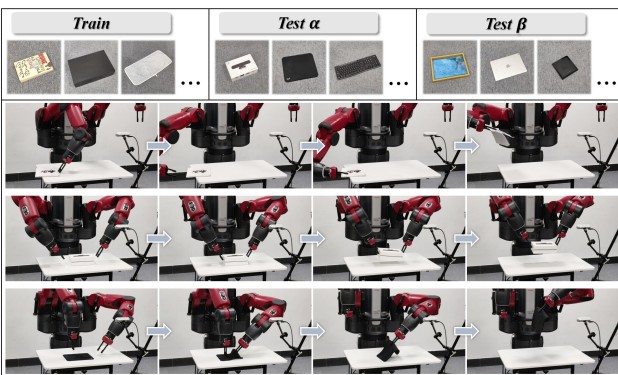

*Figure 8.* **Real-world Evaluation of Our Approach.** Experiments show that our method can successfully transfer to the real world, achieving high accuracy and strong generalization.

## 6. Conclusion and Limitation

In this paper, we tackle robotic flat object manipulation and introduce FlatLab, a comprehensive simulation platform for evaluation. We propose a unified framework that generalizes across diverse flat objects by combining multiple manipulation strategies: pushing thin items to table edges, dual-arm lifting for thick objects, and edge-squeezing for deformable sheets. Extensive experiments demonstrate the effectiveness of our approach and its strong generalization to unseen objects and categories. Several limitations remain. First, most evaluations are conducted in simulation, and bridging the sim-to-real gap is a key direction for future work. Second, the impact of increasing scene complexity on manipulation performance has yet to be systematically studied. Third, integrating large vision-language-action models into flat object manipulation presents a promising avenue for further research.

of **83.6%**, **80.0%**, and **80.0%** on the **Train**, **Test** $\alpha$, and **Test** $\beta$ sets, respectively, confirming the effectiveness of our method in the real world. A subset of the objects and the experimental workflow are illustrated in Figure 8. The physical factors in the real world are more significant. For example, rigid objects have more stable friction, and deformable objects can easily form graspable wrinkles when squeezed. Conversely, in simulation, insufficient friction may cause objects to slide off easily, and unstable deformable simulations may result in objects being ejected or indistinct wrinkles. These results where simulation performs worse than reality also confirm the greater applicability of our method in the real world, and it proves the superiority of designing manipulation strategies from the perspective of physical properties. More details are provided in Appendix H.

## Acknowledgements

This work was supported by the National Natural Science Foundation of China (No. W2421093) and the International Cooperation Project of Jilin Province (No. 20250205079GH). This work was supported in part by the National Natural Science Foundation of China (No. 62476110, No. U2341229), and the Institute for Information & Communications Technology Promotion (IITP) grant funded by the Korea government (MSIP) (No. RS-2024-00397615, Development of an automotive software platform for Software-Defined Vehicles (SDV) integrated with an AI framework required for intelligent vehicles).

## Impact Statement

This paper presents work whose goal is to advance the field of robotic manipulation. There are many potential societal consequences of our work, none of which we feel must be specifically highlighted here.

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

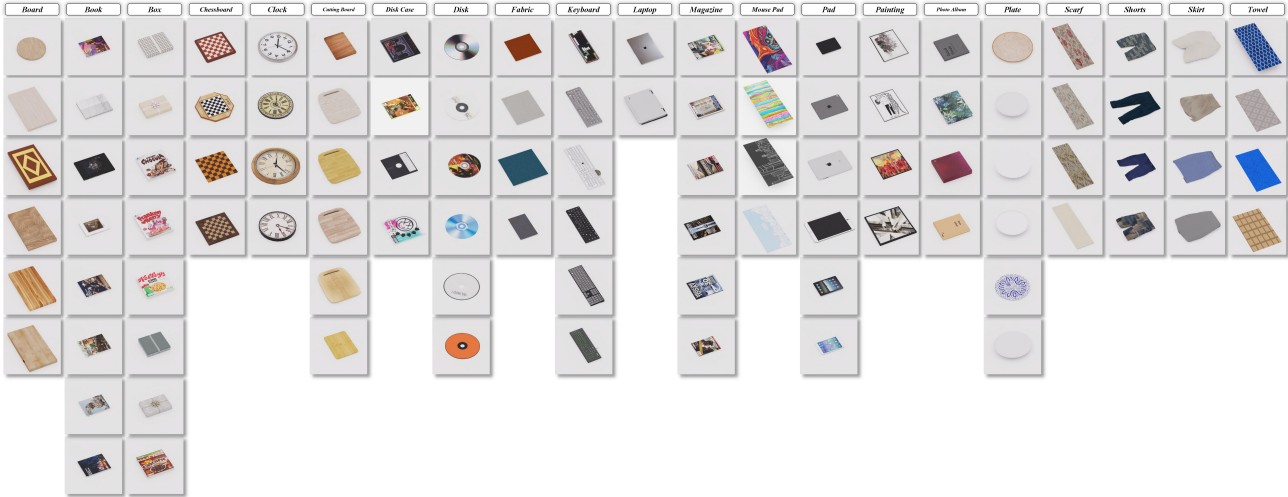

*Figure 9.* Visualization of all flat object assets in FlatLab.

# Appendix Overview

## A. Details of FlatLab

This section provides detailed implementation instructions, configuration specifications, and additional visualizations for FlatLab, with the goal of improving reproducibility and facilitating future expansion by the research community.

### A.1. Asset Library and Physical Parameterization

FlatLab includes a curated library of more than 100 flat objects spanning 21 semantic categories. All flat object assets are shown in Figure 9. FlatLab provides 18 operational scenes that are categorized into 6 classes, all of which are displayed in Figure 10. Each asset is annotated with empirically derived physical and semantic properties to enable consistent simulation and evaluation.

Each asset is associated with several explicitly specified attributes. These attributes include mesh geometry and collision shape, mass and centroid, surface friction coefficient, material appearance with texture and reflectance, semantic category, and a unique object identifier.

Resources are stored in the Universal Scene Description (USD) format, which provides a unified representation of geometry, physics, semantics, and rendering parameters. The resource catalog follows a hierarchical organization that supports scalability and extensibility. FlatLab supports both rigid objects and deformable objects. Rigid objects rely on collision and contact simulation provided by PhysX (NVIDIA, 2023), while deformable objects are simulated using Position-Based Dynamics (PBD) (NVIDIA, 2023) with GPU acceleration.

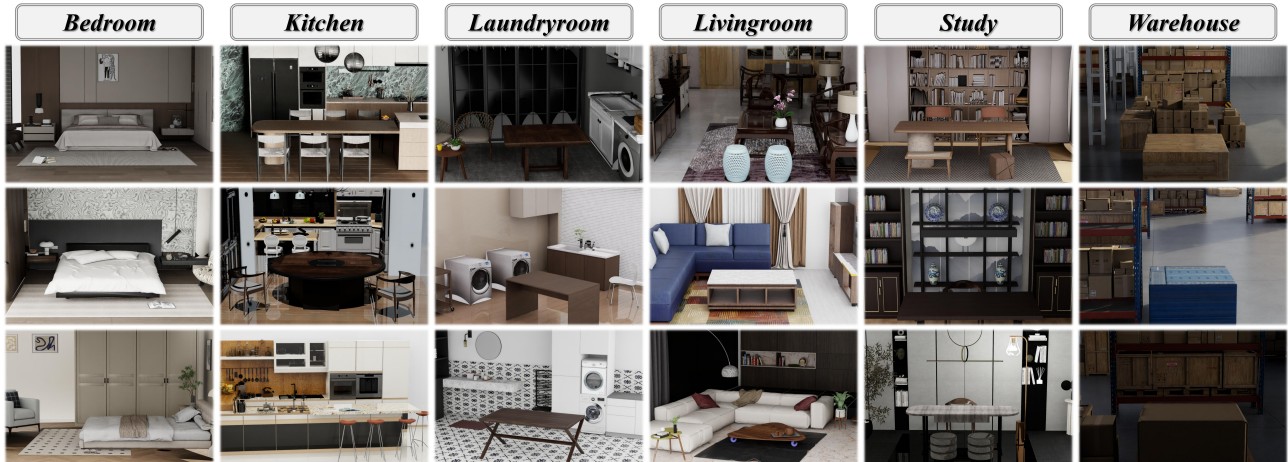

*Figure 10.* Visualization of various manipulation scenarios in FlatLab.

## A.2. Deformable Object Simulation with Position-Based Dynamics

To efficiently simulate deformable flat objects such as fabrics and towel, FlatLab employs a Position-Based Dynamics (PBD) approach that is implemented in Isaac Sim (NVIDIA, 2023). This approach represents deformable objects as particle systems subject to a series of geometric and physical constraints.

Key simulation parameters include particle resolution, which controls geometric accuracy; tensile and bending constraints, which control deformation behavior; the solver iteration count, which balances numerical stability and computational efficiency; and contact constraints that govern interactions between rigid objects and deformable objects.

All parameters are configurable at the resource level, allowing the simulation of diverse material behaviors, such as soft fabrics, moderately rigid mouse pad, or deformable towel. As a result, this design achieves stable real time simulations while maintaining sufficient physical realism, meeting the requirements of manipulative tasks such as folding, squeezing, and grasping.

## A.3. Automated Multi-Modal Data-Collection Pipeline

FlatLab provides a fully automated workflow for acquiring large scale multi-modal robotic manipulation datasets. This workflow eliminates the need for manual remote operation or developing inverse kinematics scripts.

At the start of each training phase, the environment manager randomizes the object pose, initial robot configuration, camera viewpoint, and chosen physical parameters. Subsequently, the task scheduler samples a task instance and generates a corresponding sequence of robot actions.

During execution, the data collector records multiple data modalities, including RGB images and depth maps captured from configurable multiple view cameras, point clouds reconstructed from depth observations, robot joint states, 6D poses of the end effectors, and task level annotations such as success or failure labels.

Each trial is automatically evaluated according to task specific success criteria, and the evaluation results are stored in a structured dataset. Users can select any subset of modalities according to subsequent learning objectives.

### A.3.1. AUTOMATED VISUAL DATA COLLECTION

This visual data acquisition system, implemented on the NVIDIA Isaac Sim platform, supports large scale, fully automated, and efficient data acquisition.

Object resources are organized by semantic categories, with each category corresponding to a predefined set of Universal Scene Description (USD) models along with their associated instance counts. Object categories are explicitly divided into seen and unseen groups, enabling the acquired data to be used for training and evaluation of generalization.

*Table 3.* **Comparison of Simulation Platforms for Robotic Manipulation.** Columns indicate support for flat objects (rigid R / deformable D), automated data collection, demonstration data, multiple camera views (m-Camera), multiple action primitives (m-Action), and policy execution. FlatLab is the only platform providing high-fidelity simulation of both rigid and deformable flat objects with automated multi-modal data collection, demonstrations, and full action-policy integration.

| Method | Flat Object | Data Collection | Demonstration | m-Camera | m-Action | Policy |
|---|---|---|---|---|---|---|
| SAPIEN (Xiang et al., 2020) | ✕ | Manual | ✕ | ✓ | ✓ | ✕ |
| PlasticineLab (Huang et al., 2021) | ✕ | Manual | ✕ | ✕ | ✓ | ✕ |
| FluidLab (Xian et al., 2023) | ✕ | Manual | ✕ | ✕ | ✓ | ✕ |
| GarmentLab (Lu et al., 2024) | ✕ | Manual | ✕ | ✓ | ✓ | ✕ |
| ThinShellLab (Wang et al., 2024b) | ✓ (D) | Automated | ✕ | ✕ | ✓ | ✕ |
| GraspLargeFlat (Wang & Kasaei, 2025) | ✓ (R) | Automated | ✕ | ✕ | ✕ | ✓ |
| LabUtopia (Li et al., 2025) | ✕ | Automated | ✓ | ✓ | ✓ | ✓ |
| DexGarmentLab (Wang et al., 2025b) | ✕ | Automated | ✓ | ✓ | ✓ | ✓ |
| **FlatLab (Ours)** | ✓ (R+D) | Automated | ✓ | ✓ | ✓ | ✓ |

At the start of each simulation round, scene level randomization is applied to increase data diversity. This includes planar translations and rotations around the vertical axis. For rigid objects, randomization is limited to attitude perturbations, whereas deformable objects, in addition to attitude perturbations, also exhibit shape changes generated by the initial physical simulation. After objects are placed, the simulation continues until the scene reaches a stable physical state, after which visual observation data is acquired.

Visual observation data includes RGB images and depth maps, as well as object category labels and scene-level metadata, such as object identifiers and poses. All data is stored in a hierarchical directory structure and organized by object category, scene index, and sensor view, facilitating data reproducibility and efficient loading.

### A.3.2. AUTOMATED ACTION DATA COLLECTION

The action data acquisition process aims to obtain long term operational trajectories through fully automated execution. Complex operational behaviors are broken down into multiple semantically meaningful action stages, each implemented as an independent control module. This staged design allows for explicit modeling of intermediate interactions and simplifies action parameterization.

Centralized execution scripts coordinate the sequential execution of all stages within a single simulation process. The simulation environment is not reset between stages, and the final state of each stage is directly used as the initial state for the next stage. This ensures temporal continuity and preserves the history of physical interactions during long term simulation runs.

In each stage, the robot's actions are parameterized by the target end effector's pose, direction of action, execution duration, and contact state. During execution, each simulation step records synchronized time series data, including robot state, action parameters, object state, and stage identifier. This forms a structured representation that explicitly aligns perception, action, and interaction states over time.

The collected action data is stored as sequential trajectories with stage-level annotations. This format is compatible with downstream learning paradigms such as behavior cloning, policy discrimination, and long term policy optimization. Combined with a visual data acquisition workflow, this automated action data acquisition system provides a reproducible and scalable foundation for studying perception action coupling in robot manipulation.

### A.3.3. DISCUSSION ON REPRODUCIBILITY

Both data acquisition processes are executed using script control and deterministic configuration files, ensuring the generation of reproducible data under identical simulation settings. The modular design of object assets, randomization strategies, and action phases allows the process to be readily extended to new object categories and operational tasks.

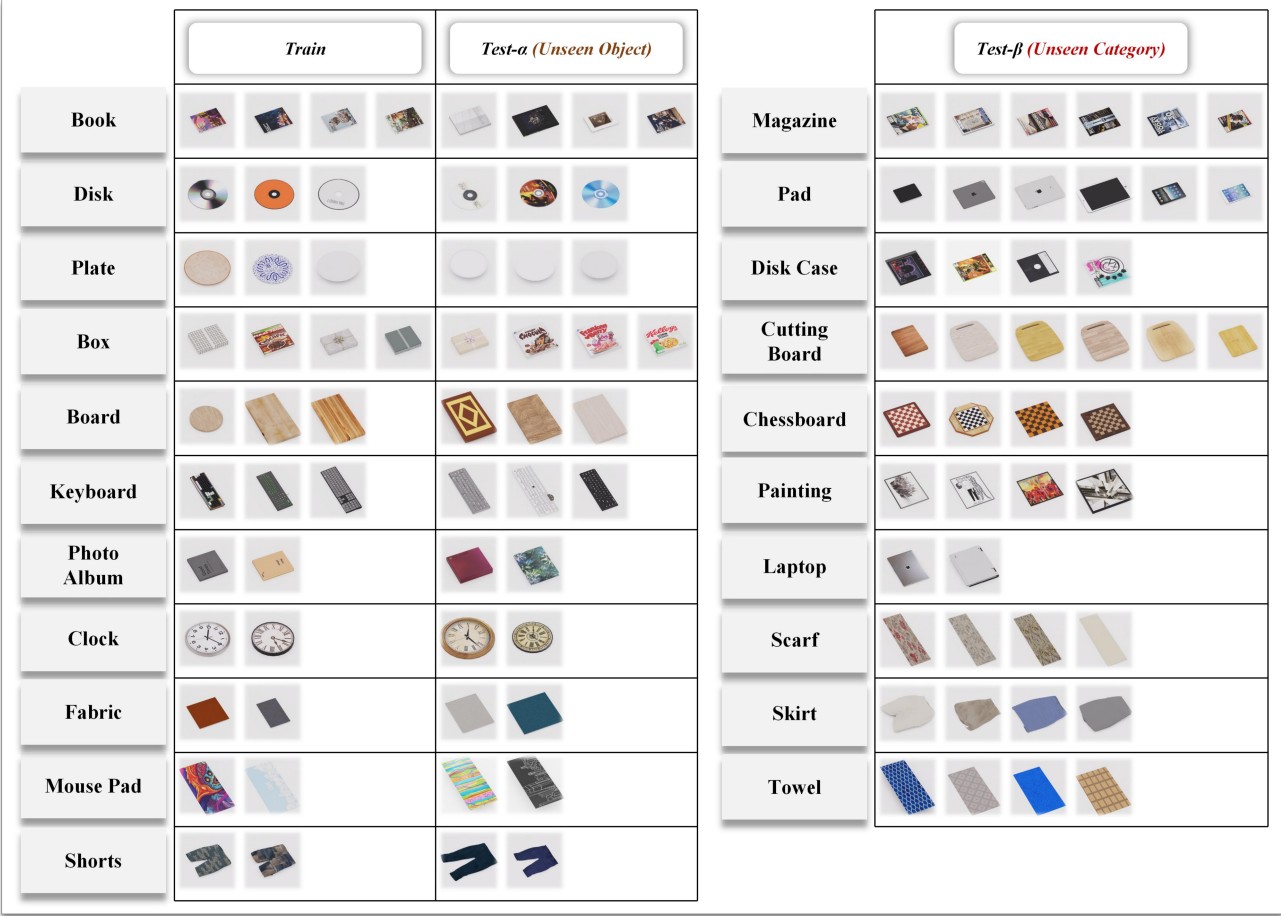

*Figure 11.* Dataset partitioning for all flat object assets in FlatLab.

## B. Comparison of Simulation Platforms for Robotic Manipulation

Table 3 compares FlatLab with representative robot manipulation simulation platforms across six key dimensions: support for flat objects (both rigid and deformable), data acquisition workflow, demonstration capabilities, multi-camera perception, motion primitives, and policy execution. As shown in the table, most existing platforms specialize in either general rigid object manipulation or isolated deformable object scenarios, leaving a systematic gap in flat object manipulation. In contrast, FlatLab uniquely integrates high-fidelity simulation for both rigid and deformable flat objects with automated multi-modal data acquisition and end-to-end policy execution capabilities within a unified framework.

## C. Dataset and Training Implementation Details

The dataset partitioning for all flat objects is shown in Figure 11. All experiments were conducted using PyTorch for model building, training, and evaluation. Unless otherwise noted, training was performed on an NVIDIA GeForce RTX 4090 GPU.

### C.1. Manipulation Strategy Generator

For the Manipulation Strategy Generator, the point cloud for each object was uniformly sampled to 2048 points, centered at zero, and normalized to lie on a unit sphere. The generator is optimized using the Adam optimizer with an initial learning rate of $5 \times 10^{-4}$ and weight decay of $1 \times 10^{-4}$. A step based learning rate scheduler is employed, reducing the learning rate by a factor of 0.5 every 30 epochs. The model is trained for 150 epochs with a batch size of 32. The checkpoint achieving the highest policy discrimination accuracy on the validation set is selected for evaluation.

*Table 4.* **Statistical analysis of experimental results on quantitative grasping of flat objects in a simulated environment.** The results are recorded as "Number of successes / Total trials". "(S)" indicates strategy discrimination, and "(G)" indicates grasping success.

| | Train (S) | Train (G) | Test $\alpha$ (S) | Test $\alpha$ (G) | | Test $\beta$ (S) | Test $\beta$ (G) |
|---|---|---|---|---|---|---|---|
| Book | 20/20 | 19/20 | 18/20 | 16/20 | Magazine | 28/30 | 25/30 |
| Disk | 15/15 | 11/15 | 15/15 | 14/15 | Pad | 22/30 | 20/30 |
| Plate | 11/15 | 10/15 | 11/15 | 10/15 | Disk Case | 17/20 | 16/20 |
| Box | 16/20 | 16/20 | 15/20 | 14/20 | Cutting Board | 20/30 | 18/30 |
| Board | 15/15 | 13/15 | 12/15 | 10/15 | Chessboard | 15/20 | 13/20 |
| Keyboard | 15/15 | 12/15 | 12/15 | 9/15 | Painting | 18/20 | 14/20 |
| Photo Album | 10/10 | 7/10 | 8/10 | 7/10 | Laptop | 6/10 | 5/10 |
| Clock | 10/10 | 8/10 | 10/10 | 8/10 | Scarf | 14/20 | 14/20 |
| Fabric | 7/10 | 7/10 | 5/10 | 5/10 | Skirt | 13/20 | 13/20 |
| Mouse Pad | 9/10 | 9/10 | 10/10 | 10/10 | Towel | 16/20 | 16/20 |
| Shorts | 10/10 | 10/10 | 8/10 | 8/10 | | | |
| **Average** | **92.1%** | **81.1%** | **82.6%** | **74.2%** | **Average** | **75.8%** | **69.0%** |

## C.2. Robot Action Execution Module

The Robot Action Execution Module is trained independently on a single GPU. Each input point cloud is uniformly sampled or zero-padded to a fixed size of $N = 8192$ points and normalized to have a mean of zero. During training, standard point cloud data augmentation methods were applied to improve robustness, including random scaling within the range of $[0.9, 1.1]$, Gaussian jitter with a standard deviation of 0.002, and random rotation around the vertical axis ($z$). During evaluation, no data augmentation was applied; deterministic point sampling was used to ensure result reproducibility. The execution module is optimized using the Adam optimizer with an initial learning rate of $1 \times 10^{-4}$ and a weight decay of $1 \times 10^{-4}$. Training is conducted for 100 epochs with a batch size of 8. A fixed random seed is used in all experiments to ensure consistency and result reproducibility.

## D. Details of Robotic Manipulation Evaluation

Two key metrics were evaluated for each object: **Strategy Discrimination (S)**, indicating whether the framework correctly selected the expected manipulation strategy; and **Grasping Success (G)**, indicating whether the robot successfully grasped the object without slipping or falling. The results are recorded in Table 4, formatted as "Number of successes / Total trials", showing results for each object in the **Train**, **Test** $\alpha$, and **Test** $\beta$ sets. The average strategy recognition rate and grasp success rate for all objects are summarized at the bottom of the table. Figure 12 illustrates the robot's operation in the main simulation scenario.

We analyzed common failure cases in robot grasping experiments. Irregularly shaped objects, such as Laptop, Painting, and Chessboard, exhibited lower grasping success rates, indicating limited generalization to diverse shapes. Photo albums occasionally could not be grasped due to insufficient contact area. Objects placed in extreme orientations or partially overlapping with other objects sometimes led to grasping misalignment. Incorrect selection of grasping strategies, for example using a lifting strategy instead of a sliding strategy, occasionally caused grasping failure, particularly when manipulating new objects in the **Test** $\beta$ set.

These failure cases suggest potential avenues for future improvement. Incorporating shape aware grasping planning or haptic feedback could increase success rates for irregular objects. Data augmentation using extreme pose configurations may reduce sensitivity to object orientation. In addition, leveraging multi-modal inputs, such as RGB-D images and point clouds, can enhance policy prediction and improve generalization to unseen object categories.

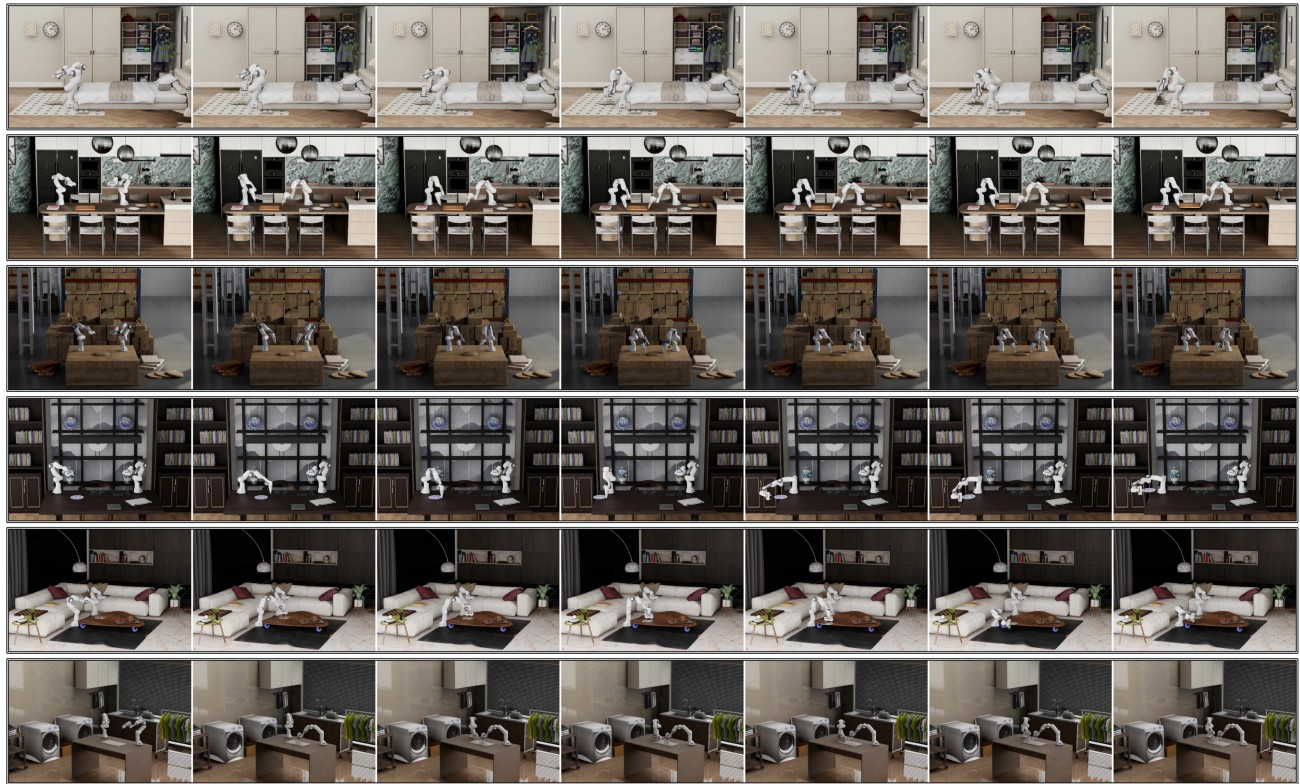

*Figure 12.* Robot flat object manipulation procedures in major simulation environment scenarios.

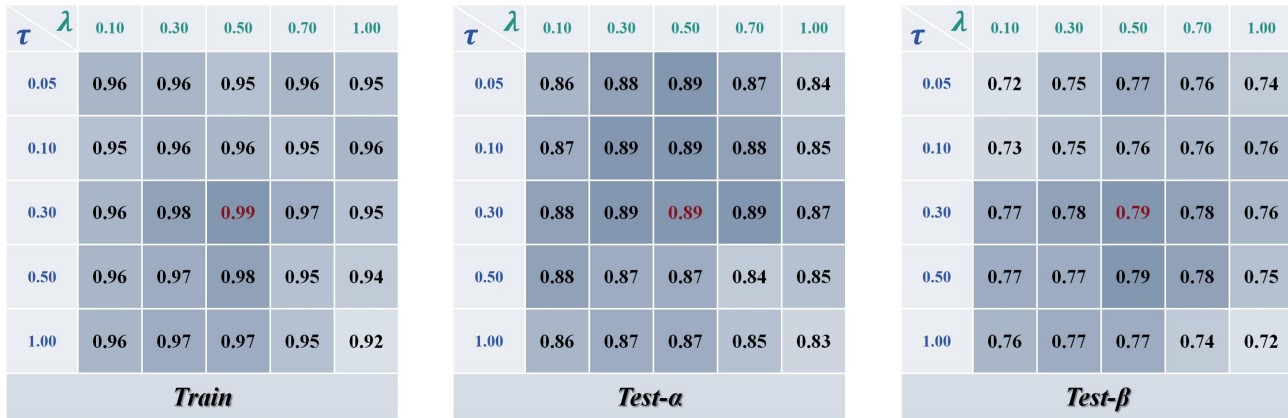

*Figure 13.* Statistical results of the strategy discrimination success rate under different settings of the hyperparameters $\lambda$ and $\tau$.

## E. Sensitivity Analysis of Method Parameters

### E.1. Manipulation Strategy Generator

We first analyze the sensitivity of the Manipulation Strategy Generator to its key hyperparameters, namely the contrastive loss weight $\lambda$ and the temperature parameter $\tau$. All experiments were evaluated on the **Train**, **Test** $\alpha$, and **Test** $\beta$ sets, and the results are summarized in Figure 13.

The weight $\lambda$ controls the relative importance of the contrastive consistency objective, which enforces strategy invariant

*Table 5.* **Sensitivity analysis of pose position loss weights in robot manipulation. P-MSE** measures the mean squared Euclidean distance between the predicted and ground truth object positions. **R-Err** measures the angular difference between the predicted and ground truth orientations.

| $\lambda_{rot} = 1.0$ | P-MSE | R-Err | P-MSE | R-Err | P-MSE | R-Err | P-MSE | R-Err | P-MSE | R-Err |
|---|---|---|---|---|---|---|---|---|---|---|
| **Test $\alpha$** | 0.0048 | 0.0061 | 0.0047 | 0.0060 | 0.0038 | 0.0060 | **0.0034** | **0.0060** | 0.0040 | 0.0061 |
| **Test $\beta$** | 0.0058 | 0.0015 | 0.0045 | 0.0012 | 0.0041 | 0.0010 | **0.0040** | **0.0010** | 0.0043 | 0.0010 |
| | $\lambda_{pos} = 1$ | | $\lambda_{pos} = 3$ | | $\lambda_{pos} = 5$ | | $\lambda_{pos} = 10$ | | $\lambda_{pos} = 20$ | |

representations across different objects. When $\lambda$ is set to a very small value, performance on the test sets drops significantly, indicating insufficient regularization for learning object invariant features. As $\lambda$ increases, generalization performance improves accordingly; however, excessively large values of $\lambda$ do not yield further gains and may slightly degrade training stability. These results suggest that the proposed strategy generator benefits from a moderate level of contrastive regularization and is not highly sensitive to the exact choice of $\lambda$ within a reasonable range.

The temperature parameter $\tau$ controls the sharpness of the similarity distribution in the contrastive objective. We observe that excessively small values of $\tau$ place undue emphasis on hard negative samples, leading to unstable optimization, whereas overly large values reduce the discriminative effect of the contrastive loss. Within a relatively broad intermediate range of $\tau$, the strategy maintains stable discriminative accuracy on both the **Test $\alpha$** and **Test $\beta$** sets, indicating that the proposed method is robust to the choice of the temperature parameter.

### E.2. Robot Action Execution Module

We further investigate the sensitivity of the Robot Action Execution Module to the loss weight parameters $\lambda_{pos}$ and $\lambda_{rot}$, which balance the supervision of grasp position and rotation prediction, respectively. Quantitative results are presented in Table 5, where position mean squared error (P-MSE) and rotation error (R-Err) are used as evaluation metrics.

In this analysis, $\lambda_{rot}$ is fixed at 1.0, while $\lambda_{pos}$ is varied over a wide range. As expected, increasing $\lambda_{pos}$ consistently reduces the position prediction error. Notably, when $\lambda_{pos}$ lies between 5 and 10, the P-MSE reaches its minimum, indicating an effective balance between position accuracy and overall optimization. Meanwhile, the rotation error remains largely unaffected by variations in $\lambda_{pos}$, demonstrating that the decoupled loss formulation effectively prevents strong interference between position and rotation learning. These results confirm that the proposed loss design enables stable and independent optimization of different pose components.

## F. Details of the Baseline Comparison Experiment

Table 6 presents the quantitative grasping evaluation results for each object using three baseline methods (**Slide**, **Lift**, **Diffusion Policy** (Chi et al., 2023)), and **3D Diffusion Policy** (Ze et al., 2024) as well as our proposed **FlatLab** method. Each entry indicates the number of successful grasps for a given object across all trials.

For the Diffusion Policy, point clouds are uniformly resampled to 8192 points using farthest point sampling. The robot platform consists of a dual-arm Franka robot with 18 degrees of freedom, including seven per arm and two for the gripper. Optimization is performed using AdamW with an initial learning rate of $1 \times 10^{-4}$, and a cosine learning rate scheduler with 500 warm-up steps is employed. Each atomic action is trained for 5000 epochs, and the model achieving the best performance on the validation set is selected for evaluation.

The results indicate that the sliding method performs reasonably well on objects that can be moved to the edge of a table, such as books and disks, but performs poorly on objects that require precise lifting or two-arm deformable extrusion manipulation, such as boxes and shorts. The lifting method shows strong performance for objects suitable for two-arm lifting, including boxes and paintings, but fails on thinner or deformable objects, such as magazines and scarves. The diffusion policy exhibits limited performance due to overfitting. In contrast, FlatLab consistently outperforms all baseline models across all datasets, demonstrating robust cross-class generalization. These findings highlight the advantage of dynamically selecting manipulation strategies based on object shape and material properties, rather than relying on fixed strategies or fully end-to-end approaches.

To further evaluate long-horizon generalization, we additionally construct a cluttered desktop clearing task in which five

*Table 6.* Comparison of quantitative experimental results for grasping evaluation in FlatLab across various baselines.

|  |  | **Slide** | **Lift** | **Diffusion Policy** | **3D Diffusion Policy** | **FlatLab (Ours)** |
|---|---|---|---|---|---|---|
| **Train** | Book | 19/20 | 10/20 | 14/20 | 13/20 | 19/20 |
|  | Disk | 12/15 | 0/15 | 6/15 | 9/15 | 11/15 |
|  | Plate | 8/15 | 0/15 | 4/15 | 7/15 | 10/15 |
|  | Box | 0/20 | 18/20 | 10/20 | 14/20 | 16/20 |
|  | Board | 1/15 | 13/15 | 12/15 | 8/15 | 13/15 |
|  | Keyboard | 3/15 | 13/15 | 6/15 | 8/15 | 12/15 |
|  | Photo Album | 4/10 | 7/10 | 3/10 | 6/10 | 7/10 |
|  | Clock | 2/10 | 7/10 | 2/10 | 7/10 | 8/10 |
|  | Fabric | 0/10 | 0/10 | 6/10 | 9/10 | 7/10 |
|  | Mouse Pad | 4/10 | 0/10 | 4/10 | 9/10 | 9/10 |
|  | Shorts | 0/10 | 0/10 | 8/10 | 7/10 | 10/10 |
| **Average** |  | 32.3% | 41.2% | 48.8% | 66.2% | **81.1%** |
| **Test $\alpha$** | Book | 16/20 | 7/20 | 3/20 | 9/20 | 16/20 |
|  | Disk | 10/20 | 0/15 | 5/15 | 7/15 | 14/15 |
|  | Plate | 5/15 | 0/15 | 4/15 | 6/15 | 10/15 |
|  | Box | 0/20 | 16/20 | 6/20 | 13/20 | 14/20 |
|  | Board | 0/15 | 11/15 | 6/15 | 7/15 | 10/15 |
|  | Keyboard | 1/15 | 10/15 | 3/15 | 8/15 | 9/15 |
|  | Photo Album | 5/10 | 7/10 | 4/10 | 5/10 | 7/10 |
|  | Clock | 2/10 | 6/10 | 3/10 | 5/10 | 8/10 |
|  | Fabric | 0/10 | 0/10 | 5/10 | 7/10 | 5/10 |
|  | Mouse Pad | 6/10 | 0/10 | 6/10 | 6/10 | 10/10 |
|  | Shorts | 0/10 | 0/10 | 6/10 | 7/10 | 8/10 |
| **Average** |  | 27.3% | 35.0% | 36.8% | 54.2% | **74.2%** |
| **Test $\beta$** | Magazine | 22/30 | 0/30 | 14/30 | 15/30 | 25/30 |
|  | Pad | 21/30 | 0/30 | 12/30 | 15/30 | 20/30 |
|  | Disk Case | 14/30 | 0/30 | 6/20 | 7/20 | 16/20 |
|  | Cutting Board | 2/30 | 20/30 | 8/30 | 11/30 | 18/30 |
|  | Chessboard | 5/20 | 12/20 | 4/20 | 8/20 | 13/20 |
|  | Painting | 3/20 | 15/20 | 3/20 | 9/20 | 14/20 |
|  | Laptop | 0/10 | 7/10 | 4/10 | 5/10 | 5/10 |
|  | Scarf | 0/20 | 0/20 | 7/20 | 12/20 | 14/20 |
|  | Skirt | 0/20 | 0/20 | 13/20 | 15/20 | 13/20 |
|  | Towel | 2/20 | 0/20 | 14/20 | 14/20 | 16/20 |
| **Average** |  | 24.7% | 27.2% | 38.8% | 51.2% | **69.0%** |

unseen flat objects from different categories in the Test $\beta$ set are randomly placed on the tabletop. For comparison, GPT-4o is integrated with the same robot action execution module and primitive library used in our framework. Experimental results show that the GPT-4o-based pipeline achieves a success rate of 36.7%, while our framework achieves 73.3%. Although GPT-4o can recognize unseen objects and generate high-level manipulation intentions, its sequential predictions are less stable in long-horizon cluttered manipulation, where small execution errors tend to accumulate across multiple interaction stages. In contrast, our strategy-centric representation and primitive-based execution provide more stable and generalized long-horizon manipulation performance.

To further evaluate the generalization and robustness of the proposed framework, we conduct additional experiments on multiple public flat-object datasets, including GraspLargeFlat (Wang & Kasaei, 2025), PreAfford (Ding et al., 2024), and AdaptPNP (Zhu et al., 2025), achieving average grasping success rates of 80.7%, 82.5%, and 84.0%, respectively. Following the same evaluation protocol, our method consistently achieves competitive performance across different object categories, demonstrating its effectiveness in handling diverse flat object manipulation scenarios beyond the training distribution.

# G. Details of Ablation Studies

We conducted extensive ablation experiments to analyze the contribution of each key component in the proposed unified grasping framework. Quantitative results are summarized in Table 2, where **A-1** to **A-8** correspond to different ablation configurations.

## G.1. Effect of Generator Input

We first investigate the impact of different input modalities on the Manipulation Strategy Generator. Specifically, we compare object-centric point clouds ($P_{obj}$), environment-aware point clouds ($P_{env}$), and RGB observations, corresponding to configurations **Full**, **A-8**, and **A-7**, respectively. Replacing $P_{obj}$ with the environment-level point cloud (**A-8**) leads to a significant drop in strategy accuracy, indicating that the inclusion of excessive background geometry weakens discriminative cues relevant to action strategy selection. Furthermore, relying solely on RGB input (**A-7**) yields the lowest accuracy across all test sets, particularly on **Test** $\beta$. This observation highlights the limitations of appearance-based cues for inferring action strategies under geometric variations and unseen object categories. Overall, these results justify our design choice of adopting object-centric point cloud representations as the primary input to the strategy generator.

## G.2. Effect of Data Transformation

We further remove the simulated data transformation module, which includes scale normalization and material-related variations applied during training. A comparison between **A-2** and **Full** shows that removing this transformation significantly degrades generalization performance. While training accuracy remains largely unchanged, disabling the data transformation leads to a pronounced drop in strategy accuracy on unseen categories. This suggests that the model tends to overfit to the geometric scale and physical properties present in the training data. In contrast, introducing the data transformation exposes the model to a broader range of geometric and physical variations, resulting in more robust and transferable strategy representations. Overall, these results demonstrate that simulated data transformation plays a critical role in improving cross-class generalization without increasing model complexity.

## G.3. Effect of Contrast Consistency

Next, we investigate the contribution of the contrastive consistency constraints by removing the contrastive loss from the strategy generator, corresponding to **A-1** and **Full**. As shown in the results, removing contrastive regularization leads to a significant performance degradation on both test sets. This observation indicates that, without contrastive consistency constraints, the model tends to rely on object-specific geometric cues for strategy discrimination, rather than learning strategy-invariant representations. In contrast, enforcing contrastive consistency across different objects executing the same strategy encourages the network to focus on functional and relational cues that are independent of object geometry. As a result, the contrastive objective substantially enhances the model's generalization capability, effectively alleviating object-level overfitting.

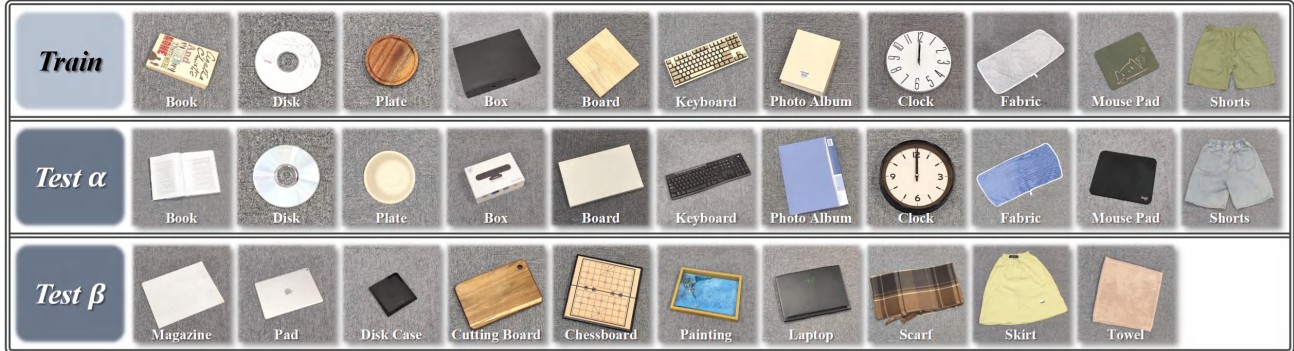

*Figure 14.* All real-world flat objects and their corresponding dataset partitioning.

### G.4. Effect of Primitive Decomposition

To evaluate the importance of primitive decomposition in the Robot Action Execution Module, we compare the proposed primitive-based execution framework with a single-stage pose regression baseline, corresponding to **Full** and **A-5**. The quantitative results demonstrate that directly regressing the grasping pose without explicit primitive decomposition leads to noticeably higher prediction errors across all evaluated settings. This suggests that a single-stage regression model struggles to capture the structured decision-making processes required for long-horizon manipulation. In contrast, decomposing the execution process into structured primitives enables the model to reason about different motion components independently, thereby reducing learning complexity and improving execution robustness. Overall, these results confirm that primitive decomposition is crucial for achieving stable and generalizable robot manipulation performance.

### G.5. Effect of Rotation Loss

Finally, we examine the impact of explicit rotation supervision in the Robot Action Execution Module by comparing configurations **A-4** and **Full**. The results show that removing the rotation loss consistently leads to higher grasping position errors. Although the position prediction task does not directly supervise object orientation, inaccurate rotation estimation adversely affects overall pose regression, resulting in suboptimal grasping configurations. This coupling effect becomes more pronounced when generalizing to unseen objects, where accurate orientation inference is critical for stable and reliable execution. By explicitly supervising rotation, the model learns a geometrically more consistent pose representation, which in turn improves position prediction accuracy. Overall, these findings confirm that rotation loss is a crucial component for achieving accurate and robust grasping pose estimation.

## H. Details of Real-World Evaluation

Our real-world flat objects and their corresponding data partitioning are illustrated in Figure 14. Data were collected and organized for a total of 32 objects across 21 categories, including 11 objects in the **Train**, 11 objects in **Test** $\alpha$, and 10 objects in **Test** $\beta$. The object category distribution was consistent with the settings of the FlatLab simulation platform. **Test** $\alpha$ and **Test** $\beta$ followed the criteria of no overlapping objects and no overlapping categories, respectively. Five manipulation experiments were conducted for each object, and all experimental initialization settings, including randomized object positions, were aligned with the simulation configurations. Detailed quantitative experimental results are presented in Table 7, and a visualization of the execution process is shown in Figure 15.

A key factor distinguishing the real-world environment from the simulation is the contact friction between the robot's end effector and the objects. Experimental results indicate that most failures were caused by difficulties in adapting to the varying friction properties of different materials. For instance, low friction between the object and the table can cause the object to slide and fall due to the inertia generated during a push. In another scenario, low friction between the end effector and the object resulted in the object sliding off less than two seconds after being lifted by the two arms working together.

To address the gap between simulation and real-world performance, a series of adjustments were implemented. For the strategy of sliding objects to the edge of the table, denser path midpoint sampling was introduced between the start and end

*Table 7.* **Statistical analysis of experimental results for quantitative grasping of flat objects in real-world experiments.** The results are recorded as "Number of successes / Total trials". "(S)" indicates strategy discrimination, and "(G)" indicates grasping success.

| | Train (S) | Train (G) | Test $\alpha$ (S) | Test $\alpha$ (G) | | Test $\beta$ (S) | Test $\beta$ (G) |
|---|---|---|---|---|---|---|---|
| Book | 5/5 | 5/5 | 5/5 | 5/5 | Magazine | 5/5 | 5/5 |
| Disk | 5/5 | 3/5 | 5/5 | 3/5 | Pad | 4/5 | 4/5 |
| Plate | 4/5 | 3/5 | 5/5 | 4/5 | Disk Case | 5/5 | 5/5 |
| Box | 5/5 | 5/5 | 4/5 | 4/5 | Cutting Board | 5/5 | 4/5 |
| Board | 5/5 | 5/5 | 5/5 | 4/5 | Chessboard | 5/5 | 3/5 |
| Keyboard | 5/5 | 3/5 | 5/5 | 4/5 | Painting | 2/5 | 2/5 |
| Photo Album | 4/5 | 4/5 | 4/5 | 3/5 | Laptop | 3/5 | 2/5 |
| Clock | 4/5 | 4/5 | 5/5 | 4/5 | Scarf | 5/5 | 5/5 |
| Fabric | 5/5 | 5/5 | 5/5 | 5/5 | Skirt | 5/5 | 5/5 |
| Mouse Pad | 5/5 | 4/5 | 5/5 | 3/5 | Towel | 5/5 | 5/5 |
| Shorts | 5/5 | 5/5 | 5/5 | 5/5 | | | |
| **Average** | **94.5%** | **83.6%** | **96.4%** | **80.0%** | **Average** | **88.0%** | **80.0%** |

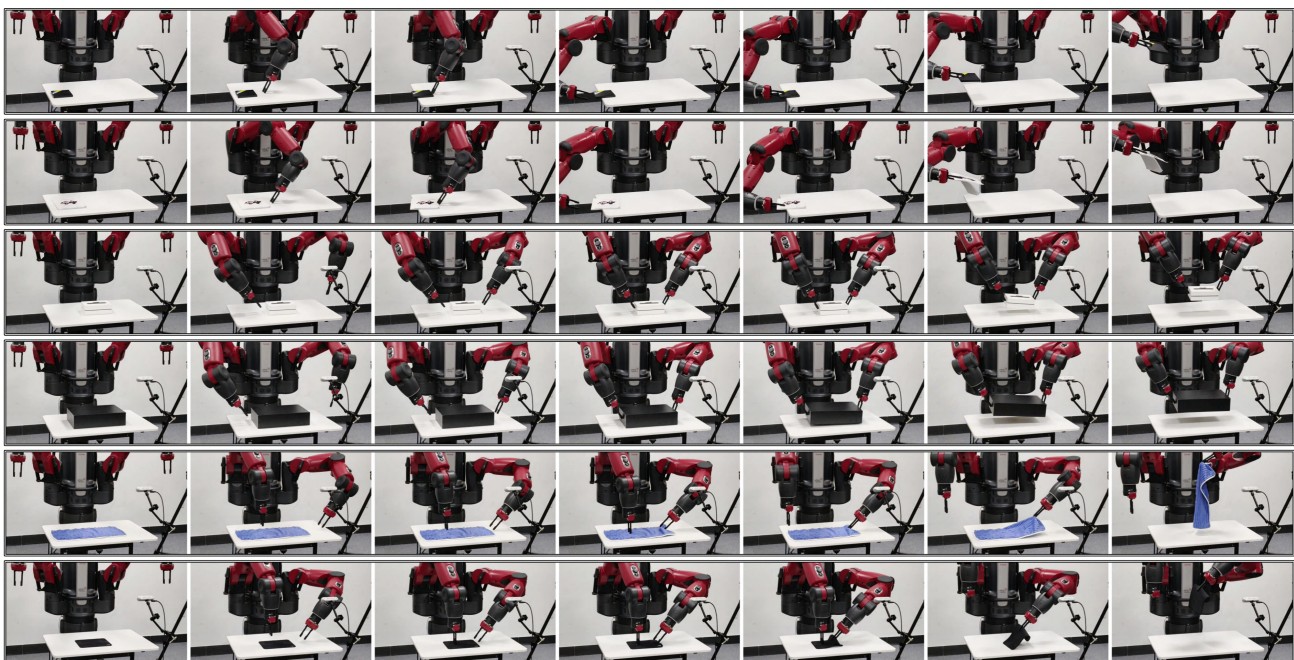

*Figure 15.* Major real-world robot manipulation procedures.

positions to reduce inertial effects. For the dual-arm cooperative lift strategy in our proposed framework, a final pose was additionally defined that continues to squeeze along the original path direction, maximizing contact with the object. Notably, this approach achieved better performance in real-world experiments than on the simulation platform for deformable objects, demonstrating the effective adaptability of our strategy to the physical characteristics of deformable objects in real-world scenarios.

