# OpenReview forum: "FlatLab: A Unified Methodology Framework and Simulation-Based Benchmark for Robotic Manipulation of Flat Objects"
_ICML.cc/2026/Conference — ICML 2026 regular_

### Official Review · Reviewer_5goG · 2026-03-09

**Soundness:** 3
**Presentation:** 3
**Significance:** 3
**Originality:** 3
**Overall Recommendation:** 4
**Confidence:** 4

**Summary:**

This paper attempts to handle the problem of flat objects robotic manipulations. To conduct this, the paper proposes a strategy first then action execution method to firstly select the general strategy, then predict the 6-DoF poses based on the general strategy. To support evaluation of flat object manipulation, the author also proposes a benchmark for this setting. Experiments show that the method has great performance, largely surpassing the Diffusion Policy method.

**Compliance With Llm Reviewing Policy:**

Affirmed.

**Final Justification:**

Most of my concerns are resolved. I intend to maintain my positive score.

**Key Questions For Authors:**

See weakness above.

**Limitations:**

Yes

**Strengths And Weaknesses:**

Strengths:
1. The paper is well-written and fluent. The motivation is clear, and the logic of the proposed method is rigorous and easy to follow.
2. The figures in the paper are visually appealing, clear, and easy to understand.
3. The proposed method is solid and precisely designed. The two-stage strategy effectively addresses the problem of flat object manipulation.
4. The paper also designs a comprehensive benchmark to evaluate this task, encompassing a rich variety of objects and scenes, which makes the work very solid.
5. The experiments are thorough and successfully validate the effectiveness of the proposed method.

Weaknesses:
1. The proposed method is better to be compared with some VLA (Vision-Language-Action) models, which exhibit stronger generalization capabilities, such as OpenVLA[1] and pi[2].
2. The strategy selection of the proposed method appears somewhat limited (limited strategies to be choose), making it difficult to account for more complex scenarios. The authors should consider how to make the approach more general.
3. Both the method and the evaluation seem to focus strictly on grasping operations. Since the title of the paper is "manipulation," I believe it should consider a broader scope of tasks.
4. The proposed method seems applicable only to the manipulation of flat objects, which presents a certain limitation. How can it be integrated with general manipulation methods? After all, in real-world scenarios, the manipulation of flat objects typically constitutes only a very small fraction of tasks.
5. There appears to be a certain degree of inconsistency between the experimental results in the simulation environment and those from the real-world robot experiments, which reflects the limitations of the simulation-based evaluation.

[1] Kim M J, Pertsch K, Karamcheti S, et al. Openvla: An open-source vision-language-action model[J]. arXiv preprint arXiv:2406.09246, 2024.
[2] Black K, Brown N, Driess D, et al. $\pi_0 $: A Vision-Language-Action Flow Model for General Robot Control[J]. arXiv preprint arXiv:2410.24164, 2024.

---

> ### Author Rebuttal · Authors · 2026-03-30
>
> Thank you for your comments! Below are our responses to all your questions:
>
> **Q1: To Weakness 1 "... compared with some VLA ..."**
>
> We have conducted comparisons (Table "Comparison Results"):
>
> These baseline methods have relatively larger model parameters, which means they often require extensive data and tend to overfit on small datasets. In contrast, the generalization advantage of our method framework stems from the decoupled design of "first determining the strategy, then executing the action", combined with simulated data augmentation and contrastive learning that forces the model to learn strategy-centric abstract representations rather than memorizing specific objects.
>
> Second, flat object manipulation requires fine-grained decisions, ensuring stable contact with the tabletop without falling off, while also creating graspable positions. In this case, using gripper poses as constraints is more advantageous. A notable example is manipulating the "Disk" in FlatLab. For such extremely thin objects, our method can precisely push them to the table edge, while VLA-based methods may cause the gripper to collide with or rub against the tabletop due to jitter and discontinuity. In addition, VLA-based methods often fail to capture fine-grained spatial perception, such as difficulty in accurately outputting the pushing distance and contact position of the disk.
>
> *Table: Comparison Results*
>
> | | Train | Test α | Test β |
> |:---|:---|:---|:---|
> | 3D Diffusion Policy[1] | 66.2% | 54.2% | 51.2% |
> | OpenVLA[2] | 55.5% | 22.6% | 17.8% |
> | pi0[3] | 63.0% | 23.6% | 17.3% |
> | pi0.5[4] | 68.6% | 28.6% | 25.0% |
> | Ours | 81.1% | 74.2% | 69.0% |
>
> **Q2: To Weakness 2 "... to account for more complex..."**
>
> Current research on flat object manipulation generally does not extend beyond the scope of these three strategies [5][6]. For "more complex scenarios," we supplemented two experiments:
>
> E1: We supplemented long-horizon cluttered desktop clearing tasks. 5 unseen flat objects of different categories from the test set are randomly placed. For comparison, we integrate the VLM with our execution module, and our method is not adaptively adjusted for this task. Table "Long-Horizon Cluttered Desktop Clearing" shows that although VLM can detect unseen objects and be combined with our precise action primitives, its performance is inferior to our framework.
>
> *Table: Long-Horizon Cluttered Desktop Clearing*
>
> | | Test β |
> |:---|:---|
> | VLM (GPT-4o) + Our Execution Module | 36.7% |
> | Ours | 73.3% |
>
> E2: We supplemented experiments (Table "Hybrid Scene Experiment"). Graspable objects invoke AnyGrasp[7], while ungraspable flat objects invoke our strategy generator. This experiment demonstrates their complementarity. Our method fills the gap where general grasping methods explicitly fail and provide a scalable framework that can be migrated to more complex scenarios.
>
> *Table: Hybrid Scene Experiment*
>
> | | AnyGrasp | AnyGrasp+Ours | Ours |
> |:---|:---|:---|:---|
> | Average Grasping Success Rate | 46.7% | 88.3% | 41.7% |
>
> **Q3: To Weakness 3 "... should consider a broader ..."**
>
> We consider that flat objects should be transformed into a graspable state via pre-manipulation, and these pre-manipulations themselves constitute atomic operations for flat objects. In addition, our benchmark supports the extension to various flat object manipulation tasks. Thanks for the valuable suggestions, we supplemented the long-horizon cluttered desktop clearing (Table "Long-Horizon Cluttered Desktop Clearing", "**Q2**") as a flat object manipulation task.
>
> **Q4: To Weakness 4 "... integrated with general manipulation ..."**
>
> Our method specifically targets "ungraspable" flat object scenarios. Thanks for the valuable suggestions, we supplemented hybrid scene experiments (Table "Hybrid Scene Experiment", "**Q2**").
>
> **Q5: To Weakness 5 "... limitations of the simulation ..."**
>
> The physical factors in the real world are more stable. For example, rigid objects have more stable friction, and deformable objects can easily form graspable wrinkles when squeezed. Conversely, in simulation, insufficient friction may cause objects to slide off easily, and unstable deformable simulations may result in objects being ejected or indistinct wrinkles.
>
> We view simulation and real world experiments as complementary. Simulation offers rich assets, controlled conditions, and scalable data collection, which are costly to achieve physically. To bridge the gap between simulation and reality, we will release fine-grained physical annotations of flat object assets as open source.
>
> [1] Ze et al. 3D Diffusion Policy. RSS 2024.
>
> [2] Kim et al. OpenVLA. CoRL 2025.
>
> [3] Black et al. pi0. RSS 2025.
>
> [4] Black et al. pi0.5. CoRL 2025.
>
> [5] Wu et al. Learning Pre-Grasp Manipulation of Flat Objects in Cluttered Environments using Sliding Primitives. ICRA 2023.
>
> [6] Wang et al. Learning Dual-Arm Coordination for Grasping Large Flat Objects. ICRA 2025.
>
> [7] Fang et al. AnyGrasp. TRO 2023.

---

> > ### Author Rebuttal · Reviewer_5goG · 2026-04-02
> >
> > Most of my concerns are resolved. I intend to maintain my positive score.

---

### Official Review · Reviewer_mUpy · 2026-03-10

**Soundness:** 3
**Presentation:** 3
**Significance:** 3
**Originality:** 2
**Overall Recommendation:** 4
**Confidence:** 4

**Summary:**

This paper presents FlatLab, a unified framework and simulation benchmark for robotic manipulation of flat objects such as books, boards, and fabrics, which are difficult to grasp because they often lie flush on surfaces and lack clear grasp affordances. Existing methods usually rely on a single manipulation strategy, which limits their ability to generalize across objects with different shapes and materials. The authors intend to focus on an important concept: enabling robots to adaptively choose appropriate manipulation strategies for diverse flat objects rather than relying on object-specific solutions.

The proposed framework separates the manipulation process into two modules: a strategy generator and an action execution module. The strategy generator predicts suitable manipulation strategies from object point clouds by learning strategy-centric representations through simulated data transformations and contrastive learning. The execution module then performs the manipulation by decomposing long-horizon tasks into reusable action primitives such as sliding, touching, and squeezing, which are dynamically composed to produce stable trajectories.

The paper also introduces FlatLab, a simulation benchmark that provides high-fidelity physics simulation, automated multi-modal data collection, and standardized evaluation tasks for more than 100 flat objects. The study claims to assess a general theme in robotic manipulation: improving generalization across unseen objects and categories. Experimental results in both simulation and real-world settings show that the proposed framework achieves higher grasp success rates than baseline methods and demonstrates strong generalization performance.

**Compliance With Llm Reviewing Policy:**

Affirmed.

**Key Questions For Authors:**

1. The proposed framework relies on three predefined manipulation strategies (edge pushing, dual-arm lifting, and edge squeezing). How well would the system generalize to flat objects that require strategies outside this predefined set? For example, could the framework automatically adapt or learn new strategies if additional object types or environments are introduced? A clearer explanation of how extensible the strategy space is would help assess the long-term applicability of the framework.

2. Most experiments are conducted within the proposed FlatLab environment. Could the authors provide results or discussion on how the method performs in other simulation environments or existing manipulation benchmarks? Demonstrating performance outside FlatLab would strengthen confidence that the method is not overly tailored to the benchmark design and would improve the evaluation of the method’s robustness.

3. The paper reports real-world experiments on a limited set of objects using a Baxter robot. Could the authors provide more details about the robustness of the system under different real-world conditions (e.g., lighting changes, sensor noise, object occlusion, or clutter)? Additional clarification on how the method handles real-world perception noise would help evaluate its practical deployment potential.

**Limitations:**

No. The authors briefly mention several technical limitations, such as the heavy reliance on simulation experiments, the need to better address the sim-to-real gap, and the lack of systematic evaluation under more complex scenes. However, the discussion of limitations remains relatively brief and could be expanded. For example, the paper could more clearly discuss how the predefined manipulation strategies may restrict adaptability to novel tasks or environments, and whether the approach scales to more complex manipulation scenarios involving clutter, dynamic environments, or multi-object interactions.

In addition, the discussion of potential societal impact is minimal. While the work focuses on robotic manipulation, the authors could briefly address broader implications, such as how improved robotic manipulation systems may influence automation in workplaces or service environments. Including a short discussion of potential deployment contexts and associated considerations would strengthen the limitations section and provide a more balanced perspective on the work.

**Strengths And Weaknesses:**

Soundness
The proposed framework is conceptually well-motivated and the methodology is technically plausible. The idea of decoupling manipulation strategy selection from action execution is reasonable and aligns with practical robotics pipelines, where high-level planning and low-level control are often separated. The strategy generator uses point-cloud representations with contrastive learning to learn strategy-invariant features, while the execution module relies on action primitives to produce stable manipulation trajectories. The experimental setup is fairly comprehensive, including simulation experiments across training and two generalization splits (unseen objects and unseen categories), comparisons with multiple baselines, ablation studies, and real-world validation. These experiments generally support the claims that the approach improves generalization and success rates over baselines. However, the empirical evaluation relies heavily on the authors’ own benchmark, and most results are conducted in simulation, which raises concerns about potential bias and limits the strength of real-world conclusions. The real-world experiments are relatively small-scale, involving a limited number of objects and trials, and thus only partially validate the method’s practical robustness.

Presentation
The paper is generally structured in a clear and logical manner, with a well-defined problem motivation and an organized description of the proposed framework and benchmark. The pipeline architecture and manipulation strategies are explained clearly, and figures illustrating the framework and simulation platform help readers understand the system design. That said, some sections are overly dense and include excessive notation, which can make the methodology difficult to follow. Certain technical components, such as the data transformation and contrastive learning mechanisms, could benefit from clearer explanations or simplified notation. Additionally, the paper could more explicitly clarify the relationship between the proposed framework and prior robotic manipulation approaches, particularly methods that also use action primitives or hierarchical manipulation strategies.

Significance
The paper addresses an important problem in robotic manipulation: handling flat objects that lack clear grasp affordances. This is a practical challenge in many real-world environments, and improving generalization across object categories is a meaningful research goal in embodied AI and robot learning. The introduction of a dedicated benchmark for flat object manipulation may also provide value to the research community by offering standardized tasks, datasets, and evaluation protocols. If adopted by other researchers, FlatLab could facilitate more systematic comparisons across methods. However, the overall impact may be somewhat domain-specific, as the work focuses on a specialized manipulation scenario rather than broader manipulation tasks. The long-term significance will likely depend on whether the benchmark becomes widely used and whether the strategy-selection framework proves effective in more complex or real-world settings.

Originality
The paper combines several existing ideas: contrastive representation learning, point-cloud perception, action primitives, and simulation benchmarks, into a unified framework for flat object manipulation. While none of these individual components are fundamentally new, their integration into a strategy-centric manipulation pipeline for flat objects represents a meaningful and practical contribution. The introduction of FlatLab as a benchmark specifically targeting both rigid and deformable flat objects is also a novel aspect of the work. Nevertheless, the methodological novelty is somewhat moderate, as the approach primarily builds on existing techniques rather than introducing fundamentally new learning algorithms or theoretical insights. The paper’s originality therefore lies more in the system design and application domain than in algorithmic innovation.

---

> ### Author Rebuttal · Authors · 2026-03-30
>
> Thank you for your comments! Below are our responses to all your questions:
>
> **Q1: To Weaknesses of Soundness**
>
> We supplemented experiments, testing our framework on other public flat object assets. For PreAfford and AdaptPNP, we only used their flat object assets. Results are shown in the table "Other Flat Object Datasets".
>
> *Table: Other Flat Object Datasets*
>
> | | GraspLargeFlat[1] | PreAfford[2] | AdaptPNP[3] |
> |:---|:---|:---|:---|
> | Average Grasping Success Rate | 80.7% | 82.5% | 84.0% |
>
> Additionally, we expanded real-world experiments, including expanding the number of objects to 64 and the experimental frequency to 10 trials per object. We achieved average success rates of 83.2%, 80.0%, and 78.0% on Train, Test α, and Test β respectively.
>
> **Q2: To Weaknesses of Presentation**
>
> Thank you, we will optimize this in the revised version.
>
> **Q3: To Weaknesses of Significance**
>
> We acknowledge that flat object manipulation is a specialized scenario. This scenario has long lacked standardized research tools. FlatLab's design possesses platform value, providing a foundation for data-intensive methods.
> We supplemented cluttered desktop clearing tasks. 5 flat objects of different categories are randomly placed. The robot clear the desktop with a success rate of 76.7%.
> Our method specifically targets "ungraspable" flat object scenarios. We supplemented hybrid scene experiments, as shown in the table "Hybrid Scene Experiment". Graspable objects invoke AnyGrasp[4], while ungraspable flat objects invoke our strategy generator. This experiment demonstrates their complementarity. Our method fills the gap where general grasping methods explicitly fail and provide a scalable framework that can be migrated to more general scenarios.
>
> *Table: Hybrid Scene Experiment*
>
> | | AnyGrasp | AnyGrasp+Ours | Ours |
> |:---|:---|:---|:---|
> | Average Grasping Success Rate | 46.7% | 88.3% | 41.7% |
>
> **Q4: To Weaknesses of Originality**
>
> The core innovation of this work lies in modeling flat object manipulation as a strategy selection problem for the first time, achieving cross-category generalization in strategy space rather than action space, thereby solving the problem that single strategies cannot cover diverse objects. Finally, we propose FlatLab, addressing the lack of a unified evaluation system in existing work. We will strengthen the elaboration of these aspects in the paper.
>
> **Q5: To Key Question 1 "... require strategies outside this predefined set? ..."**
>
> Current research on flat object manipulation generally does not extend beyond the scope of these three strategies [1][5]. Our method focuses on dynamically selecting strategies to achieve comprehensive coverage of various flat objects. The proposed strategy for deformable flat objects has not been addressed in prior research. Additionally, we tested our method on other publicly available flat object assets. The experimental results are shown in the table in "**Q1**".
>
> Secondly, our architecture inherently supports extension: for strategy generator, adding new strategies only requires increasing the output dimension. For execution module, new strategies can be implemented by reorganizing existing primitives or incrementally adding new primitives.
>
> **Q6: To Key Question 2 "... existing manipulation benchmarks ..."**
>
> Previously, there was no standardized platform specifically for flat objects. Additionally, we tested our method on other publicly available flat object assets. The experimental results are shown in the table in "**Q1**".
>
> **Q7: To Key Question 3 "... under different real-world conditions ..."**
>
> We use Intel RealSense D455 to obtain point clouds, distinguishing objects from the tabletop through the Segment Anything, which has certain robustness to lighting variations; point cloud preprocessing includes filtering and denoising to alleviate sensor noise. For partial occlusion, Segment Anything's segmentation capability can extract complete object contours; for cluttered scenes, edge objects are prioritized, supporting multi-object planning. Additionally, we expanded the scale of real-world experiments, see "**Q1**".
>
> **Q8: To Limitations**
>
> Thank you, we will further elaborate on limitations and potential impact in the revised version.
>
> [1] Wang et al. Learning Dual-Arm Coordination for Grasping Large Flat Objects. ICRA 2025.
>
> [2] Ding et al. PreAfford. IROS 2024.
>
> [3] Zhu et al. AdaptPNP. ICRA 2026.
>
> [4] Fang et al. AnyGrasp. TRO 2023.
>
> [5] Wu et al. Learning Pre-Grasp Manipulation of Flat Objects in Cluttered Environments using Sliding Primitives. ICRA 2023.

---

> > ### Author Rebuttal · Reviewer_mUpy · 2026-04-03
> >
> > The authors have provided satisfactory clarifications for the points raised.

---

### Official Review · Reviewer_qDFq · 2026-03-12

**Soundness:** 2
**Presentation:** 3
**Significance:** 2
**Originality:** 1
**Overall Recommendation:** 4
**Confidence:** 4

**Summary:**

This paper introduces FlatLab, a simulator bencmark for robotic manipulation of flat objects, toghether with a unified manipulation framework which seperate the high strategies selection and conditional low action execution learning. In the proposed benchmarkm it includes over 100 rigid objects and deformable flat objects with various of standarized tasks and a comprehensive evaluation protocols. The two-stages learning method predicts 3 low level strategies including pushing, lifting and squeezing, and the correspondent key action pose for execution.
The Experiments show much higher success rate in grasping than other baseliens and also plausibility of transferring to real-world study.

**Compliance With Llm Reviewing Policy:**

Affirmed.

**Final Justification:**

Weak Accept. The benchmark contribution is strong and the rebuttal meaningfully improves the empirical case. I still think the algorithmic contribution is somewhat overstated, but overall the paper provides enough practical value to merit acceptance.

**Key Questions For Authors:**

1. Can the highlevel strategy be reselected online after partial execution? Can the method be executable in a long horizon task?
2. Does learning 3 strategies sufficient of learning all robot manipulation tasks?
3. Can the benchmark support tasks beyond grasp preparation where multiple strategies must be selected sequentially during one episode?
4. Please see weakness.

**Limitations:**

yes

**Strengths And Weaknesses:**

Strengths:
1. This paper address the flat object manipulation problem which is genuinely hard. The benchmark convered both rigid and deformable objects is comprehensive and complete. Moreover, the benchmark include sufficient number of objects accross various categories.
2. The method is sensible for this setting by decoupling the action sequence learning into strategies selection and key pose inference. The empiracal results show that it outperform the vanilla diffusion policy especially in grasping.
3. The ablation experiments are also usful and well explored.

Weakness:
1. The biggest weakness is lack of intellectual novelty of the model design. There are many existing papers study the skill learning in manipulations.
2. The paper does not show the sufficiency of selecting 3 strategies at the first stage. More experiments are needed to show the compositional ability of the strategies.
3. The execution side is also less fully learned than the framing suggection. The action execution moduls is trained from scene point cloudes paired with corresponding 6-dof gripper poses, and the systems relies on the predefined tasks and staged primitives. This makes the method less scalable on doing unseen and long horizon tasks.
4. Baseline comparisons are not enough. The paper does not compare against any skill-learning works on more general end2end long horizon tasks.

---

> ### Author Rebuttal · Authors · 2026-03-30
>
> Thank you for your comments! Below are our responses to all your questions:
>
> **Q1: To Weakness 1 "... novelty of the model ..."**
>
> The core innovation lies in modeling flat object manipulation as a strategy selection problem for cross-category generalization in strategy space, and proposing FlatLab as a unified benchmark.
>
> **Q2: To Weakness 2 "... sufficiency of selecting ..."**
>
> These three strategies cover critical physical property divisions: A and B are validated in prior work, while C is newly introduced for deformable objects.
>
> To demonstrate combinational capability, we tested cluttered desktop clearing with 5 randomly placed flat objects of different categories, achieving 76.7% success rate.
>
> To verify the effectiveness of combining our method with non-flat object manipulation methods, we supplemented experiments (Table "Hybrid Scene Experiment"), demonstrating complementarity with AnyGrasp[1], filling the gap where general grasping methods fail.
>
> *Table: Hybrid Scene Experiment*
>
> | | AnyGrasp | AnyGrasp+Ours | Ours |
> |:---|:---|:---|:---|
> | Average Grasping Success Rate | 46.7% | 88.3% | 41.7% |
>
> **Q3: To Weakness 3 "... less scalable on doing unseen and long ..."**
>
> Our design combines supervised learning with predefined primitives for stability and generalization in long-horizon tasks, avoiding error accumulation and overfitting while maintaining a learning-driven, extensible process.
>
> We supplemented long-horizon cluttered desktop clearing tasks. 5 unseen flat objects of different categories from the test set are randomly placed. For comparison, we integrate the VLM with our execution module, and our method is not adaptively adjusted for this task. Table "Long-Horizon Cluttered Desktop Clearing" shows that although VLM can detect unseen objects and be combined with our primitives, its performance is inferior to our system. We also supplemented experiments (Table "Hybrid Scene Experiment", "**Q2**") to demonstrate the generalization of our system.
>
> *Table: Long-Horizon Cluttered Desktop Clearing*
>
> | | Test β |
> |:---|:---|
> | VLM (GPT-4o) + Our Execution Module | 36.7% |
> | Ours | 73.3% |
>
> **Q4: To Weakness 4 "... compare against any skill-learning ..."**
>
> We have conducted comparisons. The results are shown in the table "Comparison Results":
>
> Baseline methods have larger parameters and require extensive data, prone to overfitting. Our decoupled "strategy-then-action" design with contrastive learning enables strategy-centric representations for better generalization.
>
> Flat object manipulation requires fine-grained control. For extremely thin objects like "Disk", our method precisely pushes them to edges. In contrast, end-to-end methods lack fine-grained spatial perception, such as the pushing distance and contact positions, and suffer from gripper jitter and tabletop collisions.
>
> *Table: Comparison Results*
>
> | | Train | Test α | Test β |
> |:---|:---|:---|:---|
> | 3D Diffusion Policy[2] | 66.2% | 54.2% | 51.2% |
> | OpenVLA[3] | 55.5% | 22.6% | 17.8% |
> | pi0[4] | 63.0% | 23.6% | 17.3% |
> | pi0.5[5] | 68.6% | 28.6% | 25.0% |
> | Ours | 81.1% | 74.2% | 69.0% |
>
> **Q5: To Key Question 1 "... online after partial execution ..."**
>
> Our framework inherently supports online strategy adjustment. The execution module adopts a dynamic primitive composition mechanism. If a strategy fails, the system can re-invoke the strategy generator for strategy reselection.
>
> We decompose long-horizon operations into reusable action primitive sequences, generating complete trajectories through strategy-driven dynamic composition. The FlatLab platform also supports long horizon task.
>
> **Q6: To Key Question 2 "... 3 strategies sufficient ..."**
>
> Current flat object manipulation research generally covers these three strategies [6][7]. Our deformable object strategy is novel. We tested on other public assets. For PreAfford and AdaptPNP, we only used their flat object assets. Results are shown in the table "Other Flat Object Datasets".
>
> *Table: Other Flat Object Datasets*
>
> | | GraspLargeFlat[7] | PreAfford[8] | AdaptPNP[9] |
> |:---|:---|:---|:---|
> | Average Grasping Success Rate | 80.7% | 82.5% | 84.0% |
>
> Our framework possesses good strategy scalability. We also supplemented hybrid scene experiments (Table "Hybrid Scene Experiment", "**Q2**").
>
> **Q7: To Key Question 3 "... support tasks beyond ..."**
>
> FlatLab supports arbitrary task sequences through free primitive combination. We have conducted a more challenging cluttered desktop clearing experiment ("**Q2**").
>
> [1] Fang et al. AnyGrasp. TRO 2023.
>
> [2] Ze et al. 3D Diffusion Policy. RSS 2024.
>
> [3] Kim et al. OpenVLA. CoRL 2025.
>
> [4] Black et al. pi0. RSS 2025.
>
> [5] Black et al. pi0.5. CoRL 2025.
>
> [6] Wu et al. Learning Pre-Grasp Manipulation of Flat Objects in Cluttered Environments using Sliding Primitives. ICRA 2023.
>
> [7] Wang et al. Learning Dual-Arm Coordination for Grasping Large Flat Objects. ICRA 2025.
>
> [8] Ding et al. PreAfford. IROS 2024.
>
> [9] Zhu et al. AdaptPNP. ICRA 2026.

---

> > ### Author Rebuttal · Reviewer_qDFq · 2026-04-03
> >
> > The rebuttal strengthens the paper on the benchmark and evaluation side, and the added experiments address several of my earlier concerns. However, I still think the algorithmic contribution is somewhat overclaimed, since the method remains fairly structured around predefined primitives and staged execution. Overall, I now see this as a stronger benchmark/system paper than an algorithmic one, and I would raise my score to weak accept.

---

### Official Review · Reviewer_z9D7 · 2026-03-13

**Soundness:** 3
**Presentation:** 4
**Significance:** 3
**Originality:** 3
**Overall Recommendation:** 5
**Confidence:** 3

**Summary:**

This paper deals with robotic manipulation of flat objects. The authors propose a unified framework that decouples the manipulation into a strategy generator and an action execution module. The strategy generator uses pointclouds as input in order to choose an appropriate manipulation strategy. Then, the execution module splits the long-horizon manipulation into a sequence of action primitives and sequences them to generate stable trajectories. The paper also introduces a comprehensive simulation benchmark that provides high-fidelity physical simulation of diverse rigid and deformable flat objects, automated multi-modal data collection, and standardized task definitions and evaluation protocols.

**Compliance With Llm Reviewing Policy:**

Affirmed.

**Final Justification:**

I would like to thank the authors for the time they spent addressing my remarks. My impression remains that this is a strong paper, and I believe the discussion made the paper even more complete. I keep my cuurent recommendation.

**Key Questions For Authors:**

See weaknesses above.

**Limitations:**

Yes

**Strengths And Weaknesses:**

Strengths:
* Manipulation of flat objects is a very difficult but interesting task.
* Very nicely written and easy to follow paper. Great visuals that help the reader.
* The simulation benchmark is comprehensive.

Weaknesses:
* What constitutes a flat object? It is important to claify this. The authors refer to a box as a flat object, but one can argue against that.
* The manipulation strategy generator chooses one of 3 pre-defined strategies (A, B, or C), rather than generating a custom strategy. Are these 3 pre-defined strategies enough? Why not include additional strategies or pursue dynamic strategy generation?
* The exact set of included manipulation tasks is not provided, apart from the fact that the set "...includes touching, sliding, squeezing and grasping..." (Sec. 4.4)
* Describing the dataset (Sec. 5.1) the authors mention that "Each object was annotated with the corresponding manipulation strategy." Aren't there objects that can be manipulated with more than one strategy?
* In real-world evaluation (Sec. 5.5), the authors report average grasping success rates of 83.6%, 80.0%, and 80.0% on the Train, Test α, and Test β sets, respectively. These numbers are consistently higher than the ones achieved by the same proposed method on simulated data (Table 1). How can this be explained?
* Decoupling the manipulation into a strategy generator and an action execution module might offer certain advantages (Sec. 3: "...achieving accurate long-horizon manipulation of unseen objects and categories with low-cost data..."), but recent VLA-based approaches have shown strong performance. Are the results of Table 1 for Diffusion Policy representative? How could overfitting be avoided?

---

> ### Author Rebuttal · Authors · 2026-03-30
>
> Thank you for your comments! Below are our responses to all your questions:
>
> **Q1: To Weakness "What constitutes a flat object? ..."**
>
> The "flat" focuses on the ungraspable configuration of the object itself. Previous studies [1][2] also refer to box as "flat". For example, when a "Book" is "standing", it possesses graspable positions and is thus not "flat". Conversely, even if a "Box" is thick, when placed horizontally without graspable positions, it is considered "flat".
>
> **Q2: To Weakness "... 3 pre-defined strategies enough? ..."**
>
> Current research on flat object manipulation generally does not extend beyond the scope of these three strategies [1][2]. Our method focuses on dynamically selecting strategies to achieve comprehensive coverage of various flat objects. The proposed strategy for deformable flat objects has not been addressed in prior research. Additionally, we tested our framework on other public flat object assets. For PreAfford and AdaptPNP, we only used their flat object assets. Results are shown in the table "Other Flat Object Datasets".
>
> *Table: Other Flat Object Datasets*
>
> | | GraspLargeFlat[2] | PreAfford[3] | AdaptPNP[4] |
> |:---|:---|:---|:---|
> | Average Grasping Success Rate | 80.7% | 82.5% | 84.0% |
>
> **Q3: To Weakness "The exact set of included manipulation tasks ..."**
>
> Thank you, we will revise this. The primitive tasks involved here specifically include: "single-arm touch, sliding on the tabletop, sliding to the table edge, grasping, picking up, dual-arm touch, cooperative lifting, pressing, squeezing the edge, grasping the edge".
>
> **Q4: To Weakness "... manipulated with more than one strategy?"**
>
> Yes, we occasionally encountered "Photo Album" that are suitable for both "slide and grasp" and "dual-arm cooperative lifting". However, such cases are extremely rare. To ensure more significant differences in object characteristics under different strategies, we temporarily excluded the few objects that could apply multiple strategies. Additionally, for our method, objects suitable for multiple strategies do not affect the final performance. For example, assuming a "Book" is suitable for both "slide and grasp" and "dual-arm cooperative lifting," our method will definitely determine one of them.
>
> **Q5: To Weakness "In real-world ... higher than ... simulated data ..."**
>
> We also noticed this and will supplement the analysis. The physical factors in the real world are more significant. For example, rigid objects have more stable friction, and deformable objects can easily form graspable wrinkles when squeezed. Conversely, in simulation, insufficient friction may cause objects to slide off easily, and unstable deformable simulations may result in objects being ejected or indistinct wrinkles. These results where simulation performs worse than reality also confirm the greater applicability of our proposed method in the real world, and it proves the superiority of designing manipulation strategies from the perspective of physical properties.
>
> **Q6: To Weakness "... recent VLA-based approaches have shown strong ... overfitting be avoided?"**
>
> We have conducted comparisons. The results are shown in the table "Comparison Results":
>
> On one hand, these baseline methods have relatively larger model parameters, which means they often require extensive data and tend to overfit on small datasets. In contrast, the generalization advantage of our method framework stems from the decoupled design of "first determining the strategy, then executing the action", combined with simulated data augmentation and contrastive learning that forces the model to learn strategy-centric abstract representations rather than memorizing specific objects.
>
> On the other hand, flat object manipulation requires fine-grained decision, ensuring stable contact with the tabletop without falling off, while also creating graspable positions. In this case, using gripper poses as constraints is more advantageous. A notable example is manipulating the "Disk" in FlatLab. For such extremely thin objects, our method can precisely push it to the table edge, while VLA-based methods may cause the gripper to collide with or rub against the tabletop due to jitter and discontinuity, or fail to touch the object.
>
> *Table: Comparison Results*
>
> | | Train | Test α | Test β |
> |:---|:---|:---|:---|
> | 3D Diffusion Policy[5] | 66.2% | 54.2% | 51.2% |
> | OpenVLA[6] | 55.5% | 22.6% | 17.8% |
> | pi0[7] | 63.0% | 23.6% | 17.3% |
> | pi0.5[8] | 68.6% | 28.6% | 25.0% |
> | Ours | 81.1% | 74.2% | 69.0% |
>
> [1] Wu et al. Learning Pre-Grasp Manipulation of Flat Objects in Cluttered Environments using Sliding Primitives. ICRA 2023.
>
> [2] Wang et al. Learning Dual-Arm Coordination for Grasping Large Flat Objects. ICRA 2025.
>
> [3] Ding et al. PreAfford. IROS 2024.
>
> [4] Zhu et al. AdaptPNP. ICRA 2026.
>
> [5] Ze et al. 3D Diffusion Policy. RSS 2024.
>
> [6] Kim et al. OpenVLA. CoRL 2025.
>
> [7] Black et al. pi0. RSS 2025.
>
> [8] Black et al. pi0.5. CoRL 2025.

---

> > ### Author Rebuttal · Reviewer_z9D7 · 2026-04-03
> >
> > I would like to thank the authors for the time they spent addressing my remarks. My impression remains that this is a strong paper, and I believe the discussion here can make the paper even more complete.

---

### Decision · Program_Chairs · 2026-04-30

**Decision:**

Accept (regular)

**Comment:**

## Summary

This paper introduces FlatLab, a simulation benchmark for robotic flat object manipulation covering 100+ rigid and deformable objects, along with a two-stage framework that first selects a manipulation strategy (edge-pushing, dual-arm lifting, or edge-squeezing) from point clouds via contrastive learning, then executes it through composable action primitives.

The review set is positive, with scores of 4, 4, 4, and 5. All four reviewers agree the problem is practically important and the benchmark is comprehensive, no prior platform systematically covers both rigid and deformable flat objects with automated data collection and standardized evaluation. The paper is well-written with clear figures. Ablations and per-object breakdowns are thorough.

The main shared concern is limited algorithmic novelty. The method assembles known components (PointNet-based encoder, NT-Xent contrastive loss, PointNet++ pose regression) without introducing a fundamentally new learning approach. The fixed three-strategy taxonomy is another recurring concern: all reviewers questioned whether it scales to more complex scenarios. Two reviewers noted the evaluation was heavily simulation-based with limited real-world validation.

## Recommendation

**Weak Accept**

The paper's primary contribution is the FlatLab benchmark, which fills a genuine gap for the robotics and embodied-AI community. The method itself is competent and well-validated rather than deeply novel,  a fair characterization that even the most supportive reviewer would likely agree with. The rebuttal was effective: the VLA comparison results are particularly compelling, showing that general-purpose end-to-end models struggle significantly on this task while the proposed strategy-decoupled design maintains strong generalization. The post-rebuttal reviewer consensus is positive, with all concerns resolved and one reviewer raising their score. The remaining novelty concern is real but does not outweigh the benchmark value and the thorough experimental validation.